# High-resolution structures of kinesin on microtubules provide a basis for nucleotide-gated force-generation

**Zhiguo Shang[1][†], Kaifeng Zhou[1], Chen Xu[2], Roseann Csencsits[3], Jared C Cochran[4], Charles V Sindelar[1]\***

[1]Department of Molecular Biophysics and Biochemistry, Yale University, New Haven, United States; [2]Department of Biology, Brandeis University, Waltham, United States; [3]Life Sciences Division, Lawrence Berkeley National Laboratory, Berkeley, United States; [4]Department of Molecular and Cellular Biochemistry, Indiana University, Bloomington, United States

**Abstract** Microtubule-based transport by the kinesin motors, powered by ATP hydrolysis, is essential for a wide range of vital processes in eukaryotes. We obtained insight into this process by developing atomic models for no-nucleotide and ATP states of the monomeric kinesin motor domain on microtubules from cryo-EM reconstructions at 5–6 Å resolution. By comparing these models with existing X-ray structures of ADP-bound kinesin, we infer a mechanistic scheme in which microtubule attachment, mediated by a universally conserved 'linchpin' residue in kinesin (N255), triggers a clamshell opening of the nucleotide cleft and accompanying release of ADP. Binding of ATP re-closes the cleft in a manner that tightly couples to translocation of cargo, via kinesin's 'neck linker' element. These structural transitions are reminiscent of the analogous nucleotide-exchange steps in the myosin and F1-ATPase motors and inform how the two heads of a kinesin dimer 'gate' each other to promote coordinated stepping along microtubules.

**\*For correspondence:** charles.sindelar@yale.edu

**Present address:** [†]Department of Biochemistry, Brandeis University, Waltham, United States

**Competing interests:** The authors declare that no competing interests exist.

**Reviewing editor**: Anthony A Hyman, Max Planck Institute of Molecular Cell Biology and Genetics, Germany

## Introduction

Conventional kinesin is the founding member of a superfamily of molecular motors that use the energy of ATP hydrolysis to transport cargo along microtubules, serving essential roles in a wide variety of cellular processes, most notably mitosis and neuronal transport. More than 20 different members of the kinesin superfamily are identified with mitosis alone, underscoring the motor's fundamental importance in the cellular life cycle (*Rath and Kozielski, 2012*). Given the important role of this motor protein in health and disease, and particularly its growing prominence as a therapeutic drug target (*Rath and Kozielski, 2012*), it is of considerable interest to understand the basic structural and functional features of this motor in molecular detail.

Conventional kinesin (kinesin-1) dimerizes via an extended stalk domain that forms a coiled coil (*Figure 1A*), so that the two catalytic motor domains are situated at one end of the coiled-coil, while cargo-binding domains are found at the opposite end. During active motility, the dimerized motor domains take alternating, eight nanometer steps toward the microtubule plus end, tracking along single protofilaments (*Gennerich and Vale, 2009*). Underlying this behavior, each motor domain cycles between conformations that are strongly attached to the microtubule (no-nucleotide and ATP-bound) and ones that are weakly attached (ADP-bound). Additionally, binding of ATP during the microtubule-attached phase of the motor domain causes a structural element called the neck linker (*Figure 1A*) to dock along the side of the motor domain in the plus end direction (*Rice et al., 1999*). The neck linker connects the C-terminus of the motor domain to the stalk, so that docking is

**eLife digest** The inside of a cell is a dynamic environment. Large molecules such as proteins are commonly transported within a cell by 'motor proteins', which move along a network of filaments called microtubules. One group of motor proteins, the kinesins, typically have one end called a motor domain that attaches itself to a microtubule. The other end links to the cargo being carried, and a flexible 'neck' region connects the two ends of the motor protein.

Kinesins are bound together in pairs. The flexible neck region allows each motor domain in a pair to move past that of the other, allowing the kinesin to 'walk' along a microtubule in a step-like manner. Each step requires one motor domain to alternately tightly associate with, and then release from, a microtubule filament. This alternating cycle is coordinated by kinesin binding to and breaking down a molecule called ATP to form another molecule called ADP, which releases the energy needed for its next step.

This repeating cycle is possible because a motor domain changes shape when it binds to a microtubule. This shape change stimulates the release of ADP, freeing up room for a new ATP molecule to bind to the motor domain. Although relatively small, these structural changes produce larger changes in the flexible neck region that enable the individual motor domains within a kinesin pair to co-ordinate their movement and move efficiently. Many previous studies have investigated these shape changes using a technique called cryo-electron microscopy, which rapidly freezes samples and allows their structure to be recorded in high detail. However, the small size of the motor domains and their changes in shape means that this technique was not able to reveal the structures in full detail.

Shang et al. now exploit recent advances in cryo-electron microscopy to examine the structural changes of individual kinesin motor domains in greater detail. Images of motor domains bound to microtubules were made while the motor domain was in one of two different states: not bound to ATP or ADP, or bound to a chemically modified form of ATP that cannot be broken down. Shang et al. then used these images to produce models of the motor domains and compared the models with previously published images. This revealed a cleft in the kinesin motor domain that opens when it attaches to a microtubule. This cleft's 'clamshell-like' opening allows ADP to be released; it then closes when a molecule of ATP binds to it.

The opening and closing of the cleft causes the changes in the 'neck linker' of the kinesin that enable the motor protein to transport its cargo, and so links ATP binding to the movement of the motor protein. Shang et al. suggest that similar processes may also occur in other motor proteins that have not been as well studied as the kinesins.

accompanied by translocation of the cargo and the partner in the direction of travel. These nucleotide-dependent behaviors are thought to operate together to drive motility and force production.

The key structural transitions that underlie kinesin's stepping behavior remain uncertain. Kinesin possesses a Walker-type active site architecture, in which one important structural element (the P-loop) coordinates the nucleotide alpha- and beta-phosphates, while two 'switch' loops serve as nucleotide response elements (*Kull et al., 1996*; *Sablin et al., 1996*). It has long been suspected that these switch loops act as 'gamma phosphate sensors', transitioning from 'open' to 'closed' conformations in response to ATP. In this type of scheme, closure of the switch loops would propa-gate across the motor domain via a series of linked allosteric rearrangements in order to dock the neck linker to the motor domain. Moreover, because the switch II loop is N-terminally adjacent to one of kinesin's microtubule-interacting subdomains, sometimes called the switch II cluster, it was further proposed that kinesin could use the same gamma phosphate-sensing mechanism in order to control its microtubule affinity (*Vale and Milligan, 2000*; *Kikkawa et al., 2001*). Owing to more recent structural findings, however, the preceding ideas are increasingly regarded as incomplete or even incorrect (*Sindelar, 2011*). A major barrier to further progress in elucidating kinesin's mech-anism has been the inability of cryo-EM methods to much exceed nanometer resolution with kinesin-microtubule complexes (*Sindelar, 2011*), making it difficult or impossible to determine which crystal structures of kinesin (if any) correspond to the motor's functionally relevant, microtubule-attached states.

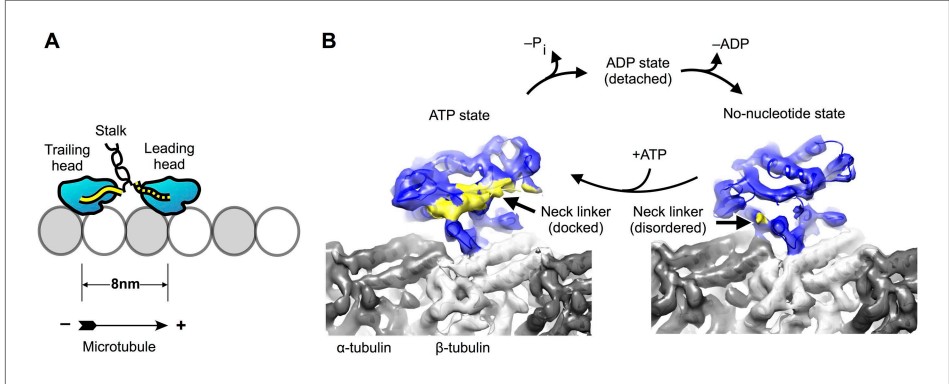

**Figure 1**. Kinesin-microtubule complex at 5–6 Å resolution. (**A**) Schematic of a stepping kinesin dimer on a microtubule protofilament. (**B**) Cross-sections of the reconstructed co-complexes of no-nucleotide kinesin (left) and ADP•Al•F$_x$ (right) with the microtubule, running parallel to a tubulin protofilament. Fourier Shell Correlation curves and other diagnostic information related to 3D refinement and reconstruction are found in *Figure 1—figure supplement 1*.
The following figure supplement is available for figure 1:

**Figure supplement 1**. Statistics and diagnostic images related to image processing and 3D reconstruction of the kinesin-microtubule complex.

Recently, a series of X-ray crystal structures of kinesin have been solved in the presence of ATP analogs that are thought to closely resemble kinesin's ATP state on microtubules; these include not only two structures of kinesin solved in the absence of protein co-factors (*Parke et al., 2010*; *Chang et al., 2013*), but also a structure of kinesin co-complexed with a dimer of tubulin (*Gigant et al., 2013*). However, it has remained unclear how closely these structures resemble the corresponding state of kinesin when bound to its physiological substrate (an intact microtubule), and structure models for other nucleotide states of microtubule-attached kinesin have remained elusive. These issues have left open the possibility that one or more of kinesin's vital functions may be governed by allosteric pathways not envisioned in the gamma phosphate sensor scheme, possibly involving twisting of the motor's central beta sheet (*Hirose et al., 2006*; *Kull and Endow, 2013*; *Arora et al., 2014*). Of particular importance is whether switch loop closure is triggered by ATP binding (as in the gamma phosphate sensor scheme) or by some other event, which would allow ATP binding to have a distinct function.

The coordination that occurs between two heads in a kinesin dimer is even more challenging to explain with existing structural models. Recent single-molecule optical trapping studies of kinesin dimers are most consistent with a model in which the leading head of a stepping dimer has substantially reduced nucleotide affinity compared with the trailing head (*Guydosh and Block, 2006*; *Clancy et al., 2011*). Related to this observation, it was also shown that the nucleotide affinity of the kinesin motor domain depends on the direction of a load externally applied to the neck linker (*Uemura and Ishiwata, 2003*). These lines of evidence indicate that kinesin's neck linker may serve as a control lever, 'gating' the affinity for nucleotide depending on whether it points forward (as in a trailing head) or backward (as in a leading head) (*Asenjo et al., 2003*; *Gennerich and Vale, 2009*). However, the precise nature of this and other proposed gating functions remains under debate (*Shastry and Hancock, 2010*) and, similar to the monomer mechanism itself, there is little consensus on the structural basis of any type of gating mechanism in dimeric kinesin (*Sindelar, 2011*).

To address these questions, we exploited recent advances in cryo-EM methodology to obtain structure models of the kinesin motor domain complexed with microtubules in the absence of nucleotide and following binding of the non-hydrolyzable transition state analog, ADP•Al•F$_x$. Analysis of these structures leads us to infer that kinesin's switch loops, controlled by microtubules, serve as a nucleotide exchange factor while ATP binding docks the neck linker via a clamshell distortion of the active-site nucleotide cleft. The corresponding allosteric pathways deduced from our structures are distinct from prior proposals that originated from X-ray structures of free kinesin or from lower-resolution cryo-EM reconstructions, including a very recent study where the reported resolution was in

the ~6–7 Å range (*Atherton et al., 2014*). Moreover, the resulting mechanistic model for the kinesin monomer can be used to explain key aspects of 'gating' between heads of a stepping kinesin dimer and provides a paradigm for future study of other motor proteins such as dynein and myosin, where atomic structures of the motor-filament complex have not yet been determined.

## Results

### High-resolution reconstructions of the kinesin-microtubule complex

In order to clarify the mechanism of kinesin's microtubule-attached cycle, we solved high-resolution cryo-EM reconstructions of the human monomeric kinesin motor domain (K349) attached to the microtubule in two different chemical states (*Figure 1B*), representing the motor before and after ATP binding (no-nucleotide and in the presence of ADP•aluminum fluoride (ADP•Al•F$_x$)). Through the application of recent advances in instrumentation and image processing, including the use of a high-speed electron-counting detector and algorithms to correct for beam-induced sample motion (*Li et al., 2013*) (See 'Materials and methods'), we were able to obtain a resolution of ~5–6 Å in both of these maps (*Figure 1—figure supplement 1A,B*), a significant increase over previous studies of the kinesin-microtubule complex (*Sindelar, 2011*; *Atherton et al., 2014*; *Goulet et al., 2014*). In some regions of the maps, the pitch of α-helices was resolved as were individual beta strands, indicating a resolution of at least 5 Å (*Figure 1—figure supplement 1F,G*). Features in the kinesin region of the map, however, appear to be limited to ~6 Å resolution, probably due to distortions of the helical lattice that could not be corrected by our image-processing algorithms. Hence, we filtered our maps to remove all signal beyond 6 Å resolution before proceeding with the analysis described below. Even after this filtering, both of our cryo-EM maps resolved all beta sheets, alpha helices, and many loops in both kinesin and tubulin as discrete entities. In particular, the paths of the motor's three principal microtubule-binding loops (L8, L11, L12) were directly resolved (*Figure 2*), which substantially aided further analysis.

### Binding of ATP analog induces a clamshell closure of the nucleotide cleft, accompanied by neck linker docking

Comparison of our no-nucleotide and ATP-state maps reveals that binding of the ATP analog causes an articulated conformational change in the kinesin domain that involves the concerted motion of three relatively well-defined subdomains (*Figure 2*). Two of these subdomains result from dividing the central beta sheet at the junction between the beta strands anchoring the P-loop and the switch II loop (beta-3 and beta-7, respectively). The third subdomain, sometimes called the switch II cluster, contains kinesin's principle microtubule interaction elements including the switch II helix and C-terminally adjacent loop L12 (*Woehlke et al., 1997*). Because these three subdomains are closely analogous to subdomains that were previously identified in the myosin motor protein, known as the upper 50kD, lower 50kD, and N-terminal subdomains (*Sweeney and Houdusse, 2010*), we have adopted a similar naming convention in current work.

Consistent with earlier cryo-EM observations (*Kikkawa et al., 2001*; *Kikkawa and Hirokawa, 2006*; *Sindelar and Downing, 2010*), the lower subdomain remains largely fixed on the microtubule surface in our maps (rotation of less than 5°). In contrast, the N-terminal subdomain makes a ~22° seesaw-like motion atop the lower subdomain (compare *Figure 2A,B* to *Figure 2D,E*) in response to binding of the ATP. This action carries the neck linker attachment site (at the C-terminus of helix alpha 6) up and away from the lower subdomain, concomitantly forming a cavity and delivering the neck linker into it. Correspondingly, in the ATP analog state, we observed density consistent with a docked conformation of the neck linker, extending towards the microtubule plus end (*Figure 1B*; *Figure 2A,B*), while in the no-nucleotide state the neck linker was evidently disordered (*Figure 1B*; *Figure 2D,E*), consistent with previous observations (*Rice et al., 1999*; *Sindelar and Downing, 2007, 2010*). Thus, seesaw movement by the N-terminal subdomain couples ATP binding to docking of the neck linker, in agreement with the conclusions of several previous cryo-EM studies that visualized nucleotide-induced transitions of kinesin at lower resolution (*Kikkawa and Hirokawa, 2006*; *Sindelar and Downing, 2007, 2010*; *Peters et al., 2010*; *Goulet et al., 2014*).

In contrast to previous analyses of these structural states of kinesin, however, our data reveal that the seesaw movement identified for the N-terminal subdomain does not extend to the other half of the beta sheet (the upper subdomain). Rigid-body fitting experiments using fragments of kinesin X-ray

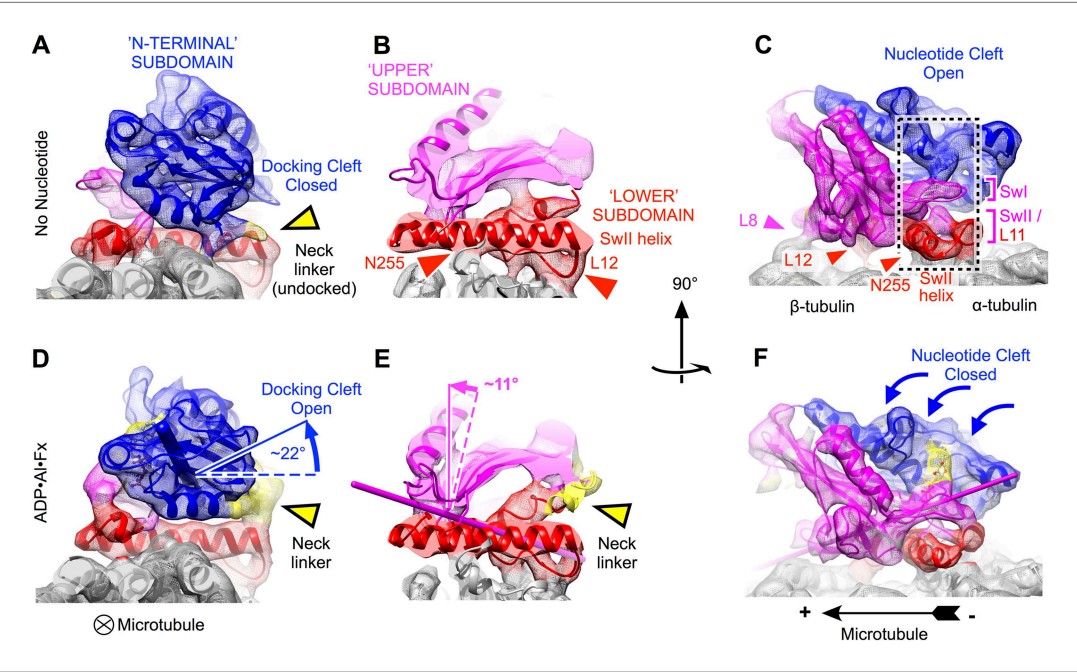

**Figure 2**. Clamshell-like closure of kinesin's nucleotide cleft triggered by ATP binding on microtubules, as revealed by cryo-EM density maps of no-nucleotide kinesin (top panels) and the ADP•Al•F$_x$ state (bottom panels). Nucleotide cleft closure is coupled, via rotation of the N-terminal subdomain, to opening of a 'docking cleft' on the opposite side of the motor domain and accompanying docking of the neck linker (**A**), (**B**), (**D**), (**E**) two different slices of density perpendicular to the microtubule axis, detailing the N-terminal and upper domains (respectively). Microtubule contacts are denoted by colored arrowheads. (**C**), (**F**) Side view of motor domain facing the nucleotide active site. Thin colored cylinders in panels **D**–**F** depict rotational axes that describe re-orientations of the N-terminal domain (blue) and upper domain (magenta) compared with the no-nucleotide state. Atomic models superposed on these maps in this and the following figures were derived from our cryo-EM data by molecular dynamics flexible fitting simulations as described in the text (see *Videos 1–3* and *Figure 2—figure supplements 1–3*).

The following figure supplements are available for figure 2:

**Figure supplement 1**. Assessing the convergence of the MDFF simulations.

**Figure supplement 2**. Improved interactions at the kinesin-tubulin interface following initial equilibration of the 4HNA starting model (see 'Materials and methods').

**Figure supplement 3**. Mobility of loop L9 (corresponding to the switch I loop) increases in simulations of unre-strained, no-nucleotide kinesin bound to microtubules, compared with simulations of the ATP state.

structures indicate that the upper subdomain rotates by much less (~11°) than the N-terminal subdomain (*Figure 2D,E*). Moreover, these fitting experiments indicate that the upper subdomain rotates about an axis that runs directly through the active site, thus predicting that the switch loops (which are anchored in the upper subdomain) would remain largely fixed in space proximal to the switch II helix as the upper subdomain rotates (*Figure 2E,F*). Related to this observation, we discovered a striking resemblance in the conformation of the switch II loop before and after binding of the ATP analog (*Figure 2C,F*; *Figure 3*). Both of our maps exhibit a contact between the switch II loop and switch II helix, coinciding with a pair of hydrogen bonds formed between the backbone of E236 and the side chain of N255 (respectively) in X-ray structures of kinesin's ATP analog-bound state (*Parke et al., 2010*; *Chang et al., 2013*; *Gigant et al., 2013*) (*Figure 3A,B*). Proximal to this contact, our maps resolve another contact between the switch II helix and the microtubule surface, corresponding to a hydrogen bond between N255 and M413 of alpha tubulin in the kinesin–tubulin co-complex structure (*Gigant et al., 2013*). Thus, our maps indicate that the upper and lower domains are pinned down against the microtubule surface by a three-way interaction between the switch II loop, the switch II

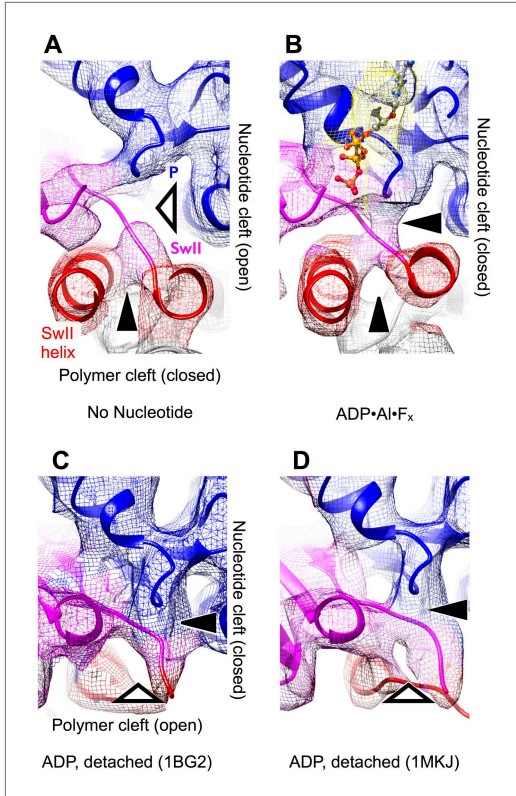

**Figure 3**. The effect of nucleotide state and microtubule attachment on kinesin's 'nucleotide' and 'polymer' clefts. Shown is an enlarged, cutaway view of the active site (rectangular region indicated in *Figure 2C,F*). Density features that define the nucleotide cleft, corresponding to the switch II loop and the P-loop, are indicated by 'SwII' and 'P' (respectively) in panel **A**. Hollow and solid wedges denote open and closed states (respectively) of the nucleotide and polymer clefts. Panels **A**–**B** show isosurface renderings our cryo-EM maps, while panels **C**–**D** show synthetic density maps generated from X-ray structures of our identical kinesin construct in its ADP-bound state, filtered to a similar resolution (6 Å) as the experimental maps in panels **A**–**B**. *Figure 3—figure supplemental 1* shows the corresponding features for synthetic 6 Å-resolution maps generated from our atomic models, as well as for two additional ADP-bound crystal structures of kinesin.

The following figure supplement is available for figure 3:

**Figure supplement 1**. Visualizing nucleotide and polymer clefts in synthetically rendered maps of kinesin at 6 Å resolution.

helix and alpha tubulin during the ATP binding transition.

A striking difference between our cryo-EM maps, however, is a distinct upward displacement of the P-loop away from the switch II loop in our no-nucleotide map. This movement appears to be a direct consequence of the 22° rotation made by the N-terminal subdomain, to which the P-loop is anchored; the rotation translates the P-loop site ~4 Å away from the microtubule surface. Consistent with this interpretation, while the P-loop forms a visible contact with the switch II loop in our ATP analog map (*Figure 3B*), a gap appears between these elements in the no-nucleotide map (*Figure 3A*). In contrast, the P-loop and switch II loop maintain very close contact in ADP-bound X-ray structures of kinesin (*Figure 3C,D* and *Figure 3—figure supplement 1C,D*). At the same time, however, the switch II loop fails to make close contact with the switch II helix in the ADP-bound X-ray structures (*Figure 3C,D* and *Figure 3—figure supplement 1C,D*). Thus, kinesin's tightly microtubule-attached states (*Figure 3*, top panels) are distinguished from the weakly attaching, ADP-bound states (*Figure 3*, bottom panels) by the presence of a gap between the switch II helix and the switch II loop; correspondingly, we will refer to this feature as the 'polymer cleft'. Similarly, the presence of a gap between the P-loop and the switch II loop correlates with the absence of bound nucleotide; we will therefore refer to this latter feature as the 'nucleotide cleft'. In the microtubule-attached states captured by our cryo-EM maps, the closed polymer cleft holds the upper and lower subdomains together while ATP binding initiates a clamshell-like action between the N-terminal domain and upper/lower subdomain assembly to close the nucleotide cleft.

## Deriving atomic models for the kinesin-microtubule complex

We used the MDFF package (*Trabuco et al., 2008*) to develop an atomic model for our ATP analog map, starting with the crystal structure of the same chemical state of the same kinesin construct co-complexed with a tubulin dimer (*Gigant et al., 2013*). This calculation employed an all-atom representation with an explicit solvation model, combined with a novel, customized steering potential derived from our cryo-EM maps (See 'Materials and methods' and *Figure 2—figure supplement 1*). In the resulting structure model, the principle conformational change observed is a straightening of the tubulin subunits from the conformation seen in 4HNA in order to accommodate the microtubule lattice (*Video 1*). Despite this change in tubulin, the conformation of kinesin is well conserved (1.08 Å backbone RMSD with the 4HNA structure for residues 9–333). Some differences with the 4HNA structure are evident at the

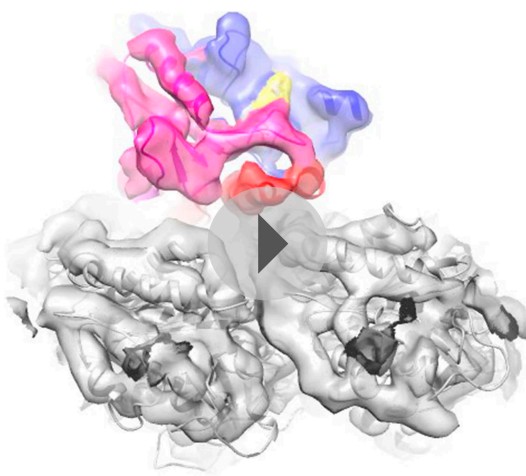

**Video 1**. Animation depicting the conformational transition observed in our MDFF simulation of the ATP bound kinesin–microtubule complex. A ribbon diagram of the molecular structure is superimposed on a semitransparent isosurface of the EM density map, filtered to 6 Å resolution. The video runs from the beginning of the MDFF simulation up to t = 1.2 ns (corresponding to the selected final model). The second half of the video plays the same transition backward. Coordinates were extracted at 50 ps intervals from the simulation trajectory. To aid in viewing, the depicted coordinate trajectory was smoothed using a 5-frame sliding window, using the 'smoothmd.py' script available from the UCSF Chimera web site (http://plato.cgl.ucsf.edu/trac/chimera/attachment/wiki/Scripts/smoothMD.py).

**Video 2**. Animation depicting our MDFF simulation for no-nucleotide kinesin, up to t = 1.4 ns.

kinesin-microtubule interface of our simulations (*Figure 2—figure supplement 2*), but these involve non-conserved side chains that re-orient to form more favorable contacts during the equilibration phase of the simulation (where the backbone conformations of kinesin and tubulin were placed under a strong harmonic constraint, prior to application of the EM steering potential). The MDFF calculation thus indicates that the conformation of kinesin visualized in the 4HNA structure is largely conserved when tubulin is incorporated into the microtubule lattice.

We repeated the above MDFF procedure to obtain a structure model for the no-nucleotide cryo-EM map. To match the chemical and structural features identified in this latter map, we removed ATP and magnesium from the active site of the 4HNA structure and deleted residues 321–349 of the neck linker, for which density in the map was weak or absent. In effect, this simulation protocol is designed to reverse the isomerization induced by ATP binding in microtubule-attached kinesin, yielding a detailed prediction for the conformation of kinesin in the absence of nucleotide (*Video 2*).

## Opening the nucleotide cleft disrupts the nucleotide binding site

The atomic models derived by the above procedure recapitulate the three-subdomain articulated movement identified by our rigid-body fitting analysis, and moreover, provide a detailed molecular explanation for the observed features at the active site. Remarkably, the switch loops remain 'closed' in an identical hydrogen-bonding network in our structural models for both nucleotide states. The hydrogen bond partners in this network, which are shared by the 4HNA starting model, extend beyond the switch loops and include not only a universally conserved salt bridge between R203 (switch I) and E236 (switch II), but also several universally conserved residues in the switch II helix (E250, N255) and L7 from the upper subdomain (Y138) (*Figure 4*, middle and lower panels). These interactions thus lock the upper and lower subdomains together, defining the closed state of the polymer cleft; the same network also locks the lower domain against the surface of alpha tubulin via a conserved hydrogen bond between N255 and the backbone of M413 from alpha tubulin.

While switch loops are thus stably maintained in a closed arrangement in these structure models, we observed a functionally significant rearrangement of the P-loop that accompanies opening of the nucleotide cleft, for the nucleotide-free model. As the P-loop pulls away from the switch loops, a

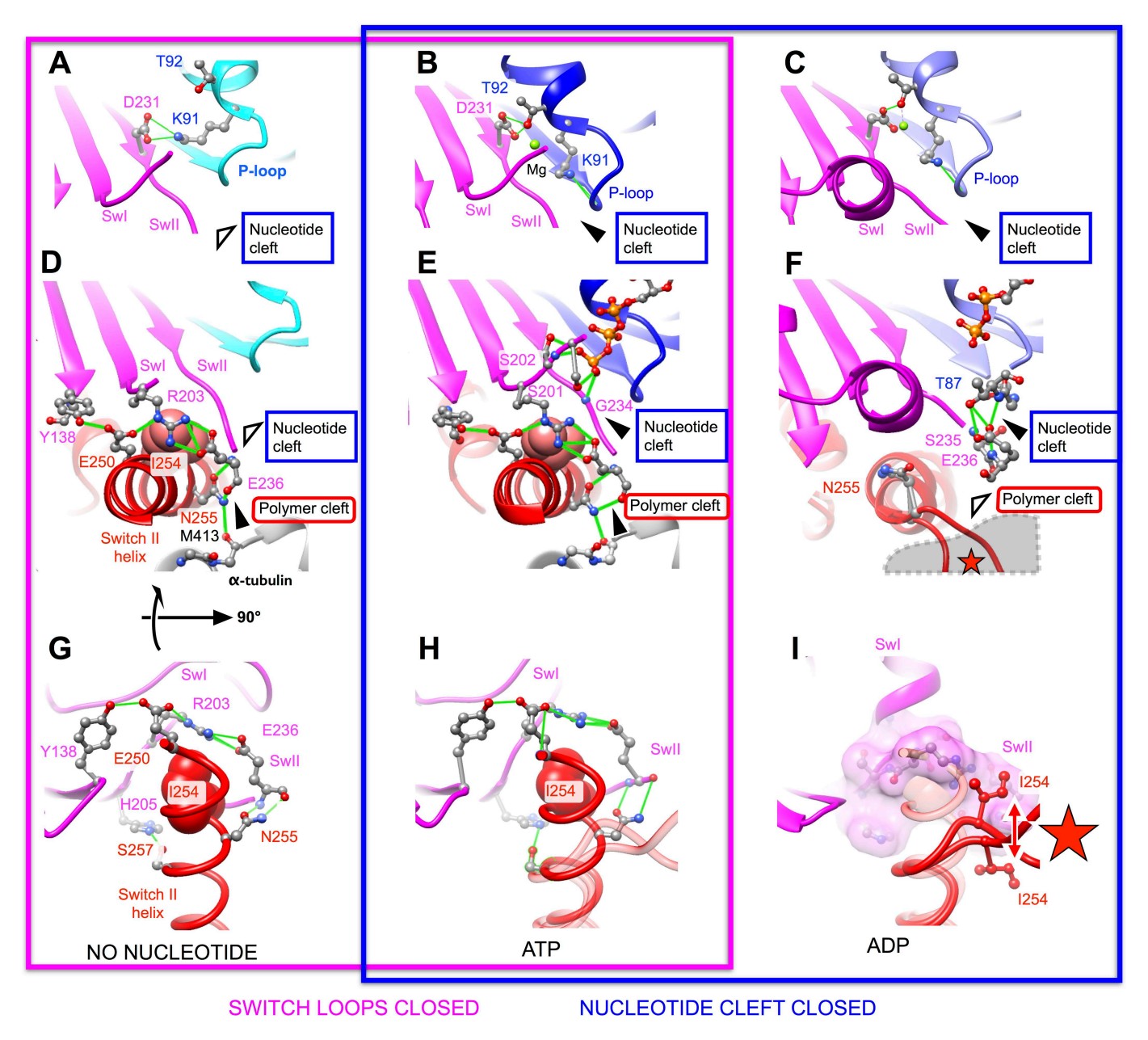

**Figure 4**. Atomic models capturing open and closed states of kinesin's nucleotide and polymer clefts for principle structural intermediates in the kinesin cycle. No-nucleotide and ATP-bound models of the kinesin-microtubule complex (this work) are depicted in panels **A**,**D**,**G** and **B**,**E**,**H** (respectively). A pair of X-ray crystal structures of the ADP-bound kinesin K349 construct (PDB ID: 1BG2, 1MKJ, which have undocked and docked neck linkers, respectively) are superposed in **C**, **F** and **I**. Panels **G**–**I** show rotated views of the structures, presenting the view from the microtubule interior looking outward at the kinesin interface. Universally conserved residues involved in the depicted structural transitions are rendered as ball and stick figures, and hydrogen bonds are depicted as solid green lines. Hollow and solid wedges denote open and closed states (respectively) of the nucleotide cleft and the polymer cleft, while the star in panels **F** and **I** indicates the site of predicted steric clashes between the switch II helix and alpha tubulin. For comparison, views of the active sites of the F1-ATPase and myosin motors in no-nucleotide and ATP analog-bound states, corresponding to panels **A** and **B**, are shown in *Figure 4—figure supplement 1*.

The following figure supplement is available for figure 4:

**Figure supplement 1**. Nucleotide cleft closure induced by ATP analogs in the myosin and F$_1$-ATPase motor proteins (compare with *Figure 4A–C*).

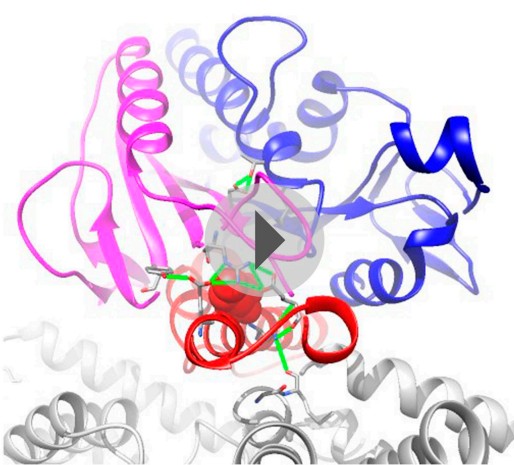

**Video 3**. Alternative depiction of the transition presented in **Video 2**, detailing the side chains and hydrogen bonds that compose the closed switch loop network (Y138, R203, E236, E250, N255), as well as residues P-loop residues K91, T92, and their interacting partner D231 from the switch II loop (compare with **Figure 3**). Hydrogen bonds formed at the beginning and endpoints of the simulation are depicted with green lines. Note that in this video the position and orientation of the outer surface of tubulin (helices H11-H12) is held fixed. In order to facilitate looping presentations of the video, the second half of the video presents the same frames of the MDFF trajectory but in reverse order.

bifurcated hydrogen bond breaks up between T92 at the base of the P-loop and S202/D231 in switches I and II (**Figure 4A,B** and **Video 3**). This action substantially perturbs the magnesium binding site, which is primarily formed by T92. A second residue in the P-loop affected by its translation away from the switch loops is K91, which normally coordinates the nucleotide alpha- and beta-phosphate groups. In the nucleotide-free structure, K91 breaks two hydrogen bonds with a pair of backbone carbonyl oxygens within the interior of the P-loop and swivels around to interact with D231 in switch II, replacing the hydrogen bond this latter residue formerly made with T92. In doing so, K91 loses its ability to coordinate the nucleotide alpha- and beta-phosphates. The resulting side chain configuration within the active site of our nucleotide-free kinesin model is strikingly similar to what is seen in X-ray structures of nucleotide-free F1-ATPase and myosin motors, which possess homologous active-site architectures (**Figure 4—figure supplement 1**). Thus, opening of the nucleotide cleft in our no-nucleotide model specifically disrupts the coordination of both nucleotide phosphate and magnesium within the P-loop.

## Open states of the polymer cleft prevent kinesin from achieving a favorable binding orientation on the microtubule surface

In order to identify the basis for microtubule-affinity regulation of kinesin by ADP and vice versa, we compared our structures of microtubule-attached kinesin with two X-ray structures (PDB ID's 1MKJ, 1BG2) of the same construct in an ADP-bound, weakly attaching state (**Kull et al., 1996**; **Sindelar et al., 2002**). The relative arrangements of the upper and lower domains in the 1BG2 and 1MKJ structures, corresponding to undocked and docked states of the neck linker, closely align with our no-nucleotide and ATP state kinesin models (respectively). We discovered, however, that both the 1BG2 and 1MKJ structures are prohibited from assuming favorable binding orientations on the microtubule surface due to the conformation of residue N255. As illustrated in **Figure 4D–F**, the N-terminal coils of the switch II helix are unwound in both ADP-bound structures. Consequently, the peptide backbone at residue N255 would point directly at the H11–H12 loop of alpha tubulin, generating a substantial steric overlap (**Figure 4F,I**). Thus, the 1BG2 and 1MKJ conformations appear incapable of strong microtubule attachment, consistent with the weak experimentally observed microtubule affinity of kinesin's ADP state (**Rosenfeld et al., 1996**).

Further comparison of our structure models to other atomic structures of ADP-bound kinesin (**Figure 5**) consistently demonstrates a clash between alpha tubulin and residue N255 (and/or adjacent residues in the switch II helix), for the ADP-state structures. While ADP-state crystal structures of kinesin are largely similar in their active-site features to the 1BG2 and 1MKJ structures, some of these exhibit an alpha-helical conformation for N255 and adjacent residues that N-terminally extends the switch II helix. However, as illustrated in **Figure 5B**, when these latter conformations of kinesin are placed in a tight-binding orientation on the microtubule (as defined by our no-nucleotide or ATP state models), the N-terminus of the switch II helix projects downward into the microtubule surface, introducing a clash between N255 and the H11-H12 loop of alpha tubulin. A recently solved 'near-rigor' conformation of the KIF14 motor (PDB 4OZQ) (**Arora et al., 2014**), found in a state that binds ADP but not magnesium, partially repairs the clash in **Figure 5B** by re-orienting the switch II helix to nearly the identical position (relative to the upper domain) as seen in our no-nucleotide model (**Figure 5C**).

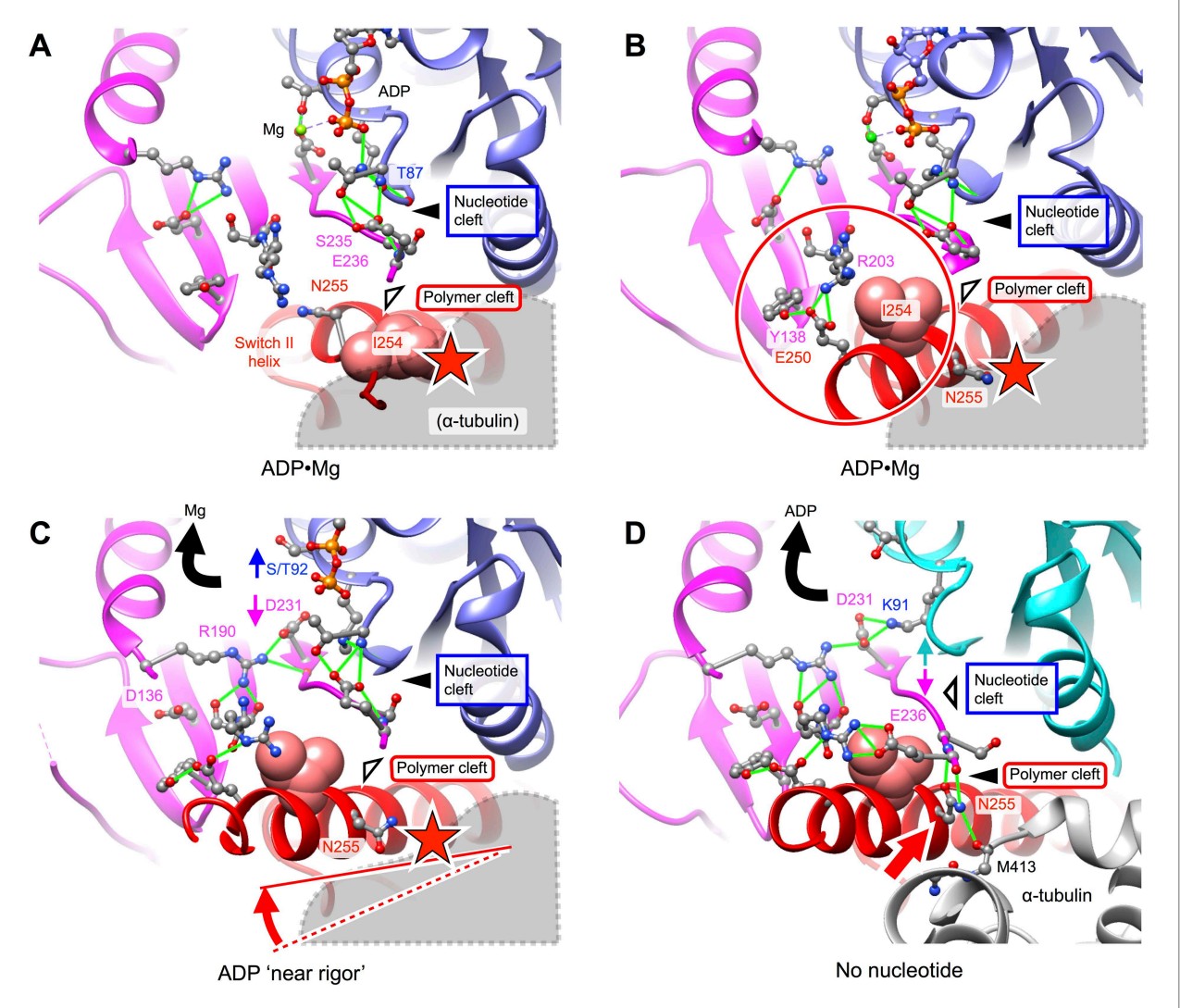

**Figure 5**. Putative intermediate states that lead to opening of kinesin's nucleotide cleft, accompanied by ADP release, upon microtubule attachment. (**A**) X-ray structure of Mg•ADP-bound K349 kinesin (PDB ID 1BG2). (**B**) X-ray structure of Mg•ADP-bound KIF1A kinesin (PDB ID: 1I5S). (**C**) X-ray structure of 'near rigor', ADP-bound KIF14 kinesin (PDB ID: 4OZQ) (**D**) No-nucleotide kinesin-microtubule complex (this work). Structures in panels **A**–**C** are aligned with the no-nucleotide K349 model in panel D by strand b7 of the central beta sheet, which connects to the switch II loop; this alignment approximately maintains the orientation of the upper subdomain, thus preserving a contact between L8 within this subdomain and alpha tubulin (see *Figure 2*). In panels **A**–**C**, predicted steric clashes between alpha tubulin and residues corresponding to I254 and/or N255 (K349 numbering) are marked with a red star.

This reorientation of the switch II helix is accompanied by inward movement of the universally conserved R591 side chain (equivalent to R190 in our construct) from loop L7, apparently dislodging S489 from D638 (equivalent to T92 and D231, respectively) to disrupt magnesium coordination (*Arora et al., 2014*), thus partially mimicking changes observed at the active site of our no-nucleotide structure. However, the side chain of N662 in the 4OZQ structure (equivalent to N255) projects downward toward the microtubule surface, introducing a steric clash with alpha tubulin.

Further comparison of the 4OZQ structure to our no-nucleotide model (*Figure 5D*) reveals that, while the overall arrangement of the N-terminal, upper, and lower subdomains is nearly identical, reorientation of N255 in the no-nucleotide structure is accompanied by striking shift of the switch II loop toward this residue and away from the P-loop. This shift opens the nucleotide cleft and closes the polymer cleft (See also *Figure 3—figure supplement 1D*) and allows the switch II loop to assume its closed conformation that simultaneously interacts with N255 as well as R203 from switch I. The

conformation of the N255 side chain that accompanies closure of the switch II loop is highly constrained by a pair of hydrogen bonds to the backbone of E236, aligning N255 in order to interact with M413 of alpha tubulin (*Figure 4G,H*). Thus, closure of the polymer cleft in our no-nucleotide model appears to be closely linked to favorable microtubule interactions by N255. Conversely, as indicated in *Figure 4F,I* and *Figure 5A–C*, the open arrangement of the polymer cleft in kinesin's ADP states is evidently inconsistent with tight microtubule attachment, owing to steric interference of N255 and neighboring residues.

## Closure of the nucleotide cleft is coupled to distortion of the peptide backbone

Previously, it has been speculated that release of nucleotide in kinesin may be linked to twisting of the central beta sheet (*Gigant et al., 2013*; *Kull and Endow, 2013*; *Arora et al., 2014*). In contrast to this expectation, however, we discovered that the degree of end-to-end twisting in both our no-nucleotide model as well as the previously reported 'rigor-like' ADP complex of KIF14 falls within the range observed in other X-ray structures of ADP-bound kinesin (*Videos 4–6*). However, we identified another mode of distortion in our kinesin models, analogous to the action of an archery bow (*Figure 6* and *Videos 7,8*), that would couple strong nucleotide binding to elastic strain in the peptide backbone. In the conformation, we identify as 'relaxed', corresponding to the no-nucleotide state, the P-loop withdraws from the switch II loop to open the nucleotide cleft. In order for the P-loop to move back toward switch II and close the nucleotide cleft, the N-terminus of alpha 2 (to which the P-loop attaches) must move with it. However, because the C-terminus of alpha 2 is covalently connected to the opposite end of the beta sheet, movement of the P-loop therefore stretches alpha 2 across the motor domain, simultaneously deforming the plus end tip of the beta sheet (*Figure 6A,B,D,E*, *Figure 6—figure supplement 1A,B* and *Videos 7 and 8*) as well as the path of alpha 2 itself (*Figure 6C*).

Comparison of kinesin's active site in various nucleotide states indicates that there are two ways to keep the nucleotide cleft closed, depending on the conformation of the switch loops. If the switch loops are open, corresponding to kinesin's detached ADP states, the switch II loop and the P-loop can interact closely with each other, which would secure the closed conformation of the nucleotide cleft (*Figure 4F*). Alternatively, when the switch loops are closed, closure of the nucleotide cleft is mainly supported by the gamma phosphate group of ATP itself, which introduces a bridge between the P-loop and the switch II loop that would hold these elements together (*Figure 4E*). In this way, our structure models indicate that tight ADP binding would be an inherent property of detached kinesin, where the switch loops are free to adopt an open conformation. However, closure of the switch loops (as accompanies microtubule attachment in our structure models) eliminates the ability of switch II to latch onto the P-loop, which would thus allow the nucleotide cleft to relax into its open conformation unless gamma phosphate is present.

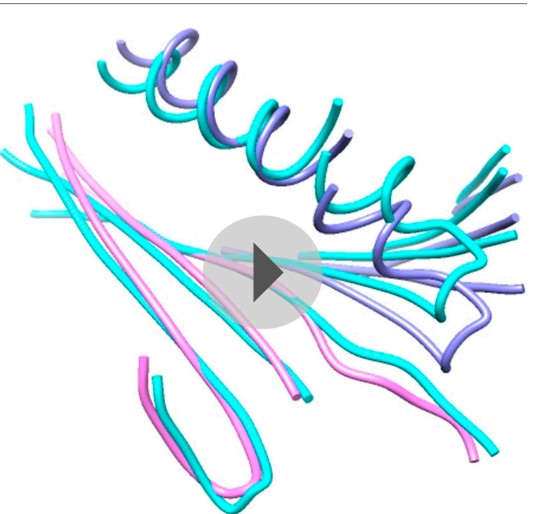

**Video 4.** Comparison of beta-sheet twist in our no-nucleotide model (cyan) and five representative X-ray structures of ADP-bound kinesin (pale magenta/pale blue). Structures in this and all subsequent videos are aligned by residues 226–231 of our kinesin model, as in *Figure 6A*. The video cycles sequentially through the ADP structures in order of increasing twist: frame 1, ADP-bound K349 (PDB ID:1MKJ); frame 2, ADP-bound KIF1A (PDB ID:1I5S); frame 3, ADP-bound K349 (1BG2); frame 4, ADP-bound KAR3 (PDB ID:1F9T); frame 5, ADP-bound Eg5 (PDB ID:1II6).

## The N255K point mutation causes loss of orientational stability on microtubules

The preceding analysis has indicated that residue N255 in kinesin plays a key role in the motor's microtubule response pathway. In effect, this site appears to act as a 'linchpin' that links closure of kinesin's switch loops to interactions at a single site on alpha tubulin (See *Figure 4D–F*). A lysine point mutation at the N255 site was previously

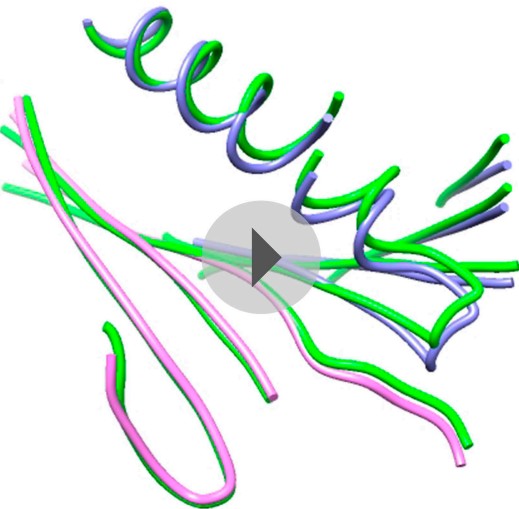

**Video 5**. Similar to **Video 4**, but substituting our no-nucleotide model with the X-ray structure of 'rigor-like' kinesin, in green (PDB ID: 4OZQ).

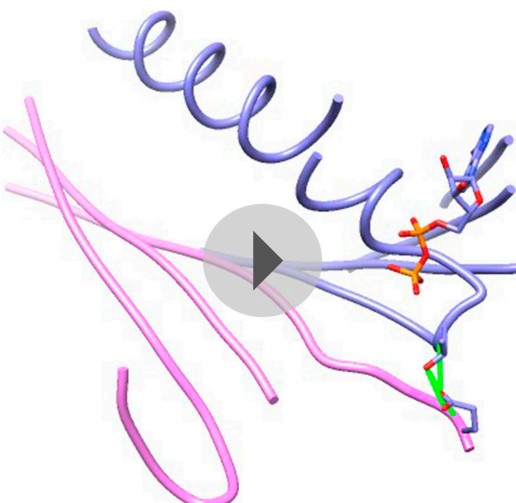

**Video 6**. Morphing animation illustrating minimal beta-sheet distortion in a comparison of ADP-bound structures of KIF1A (PDB ID:1I5S) and 'rigor-like' KIF14 (lacking magnesium; PDB ID:4OZQ). The starting structure is that of KIF14. The hydrogen bond network between residues T87 and E236 (K349 numbering), which defines a closed state of the nucleotide cleft, is depicted and the involved side chains rendered as stick figures.

shown to eliminate coupling between microtubule attachment and motor nucleotide state, allowing kinesin to bind tightly to the microtubule in all nucleotide states (**Song and Endow, 1998**; **Yun et al., 2001**; **Auerbach and Johnson, 2005**). Moreover, the microtubule-stimulated ADP release rate for this point mutant is severely attenuated, falling to only slightly above the basal level (**Auerbach and Johnson, 2005**) (**Figure 7A,B**). Based on the structure models presented here, we hypothesized that the N255K mutation uncouples nucleotide- and microtubule-binding functions in kinesin by compromising the ability of this residue to bridge between alpha tubulin and the backbone of E236 (see **Figure 4D,E**), so eliminating communication between the microtubule and the switch loops.

Cryo-EM analysis of the ADP and ATP analog-bound states of the N255K point mutant of our construct support this hypothesis, revealing two previously unidentified binding modes of kinesin in which the motor domain attaches tightly to the microtubule but is rotationally mobile (**Figure 7C–E** and **Figure 7—figure supplement 1**). The corresponding 3D maps did not refine to high resolution, presumably due to mobility of the kinesin heads. However, density for the motor domains in the N255K maps clearly indicates that a wild-type binding orientation is not achieved in either case. In the ADP-state map, the microtubule contact at site 255 appears to be attenuated or disordered (**Figure 7D**, right panel), while in the AMPPNP map the wild-type microtubule contact of loop L12 is absent but a contact is evident at the 255 site (**Figure 7E**, right panel). Thus, in neither nucleotide state is the N255K construct able to form all three wild-type microtubule contacts.

These results indicate that the specific bridging role of N255 between the switch II loop and N413 of alpha tubulin (see **Figure 4**), which likely cannot be sustained by the N255K mutation, is not required for tight microtubule attachment. On the other hand, the N255K mutant appears to be incapable of maintaining a fixed orientation on the microtubule surface. We therefore propose that N255 acts as a conformational specificity selector, forcing the switch loops to close once kinesin contacts the microtubule. While the K255 substitution lacks the appropriate geometry to fulfill this selector role, the amino group of K255 would likely form non-specific interactions with the negatively charged microtubule surface. Presumably, flexibility in the resulting contact allows the N255K mutant to maintain affinity for microtubules even when the switch loops are open. However, failure of this mutant to close the switch loops under these conditions would lead to the loss of microtubule-stimulated ADP release and catalysis, consistent with experimental measurements.

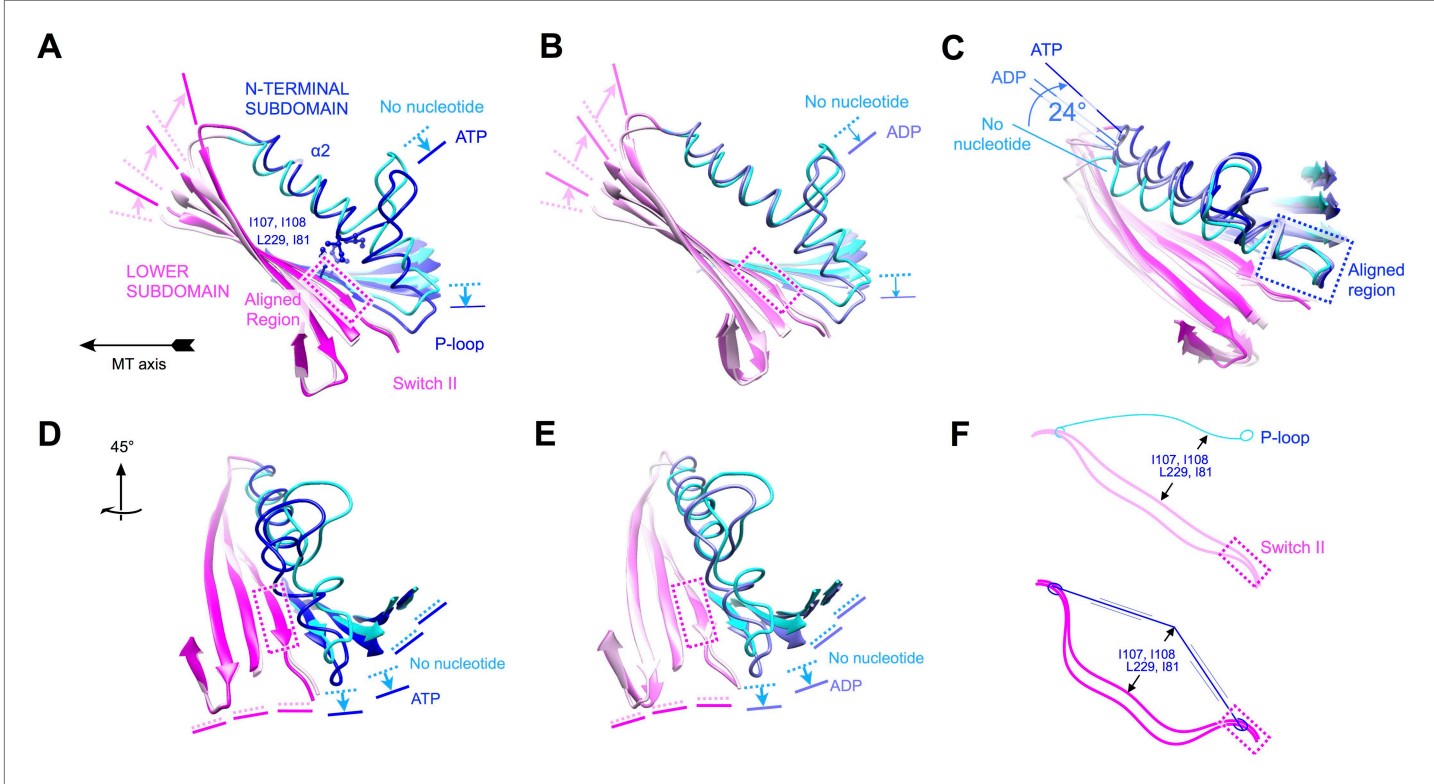

**Figure 6**. Evidence that nucleotide binding introduces internal strain in kinesin's central beta sheet and helix alpha 2, mediated by interactions between the P-loop and the switch II loop. Similar comparisons of additional structural states of kinesin are shown in *Figure 6—figure supplement 1* and *Videos 4–8*. (**A**) Flexing of the beta sheet coupled with motion of alpha 2 in comparisons of no-nucleotide and ATP-bound conformations of microtubule-attached kinesin (this work). Structures are aligned by central region of the beta strand adjoining the switch II loop (residues 226–231). (**B**) Comparison of no-nucleotide and ADP-bound (1MKJ) structures of kinesin, aligned as in panel **A**. (**C**) Comparison of no-nucleotide (current work), ADP-bound (1BG2 and 1MKJ), and ATP-bound (current work) states of the K349 kinesin construct aligned by their P-loops, revealing nucleotide-dependent flexing of helix alpha 2. (**D**), (**E**) Rotated views of panel **A** and **B**, respectively, highlighting distortion in the beta sheet proximal to the P-loop. (**F**) Cartoon depicting the inferred energy storage mechanism, by analogy to stringing an archery bow. The strain appears to be magnified by a cluster of conserved side chains (I107, I108, L229, I81; see panel **A**) that separate alpha 2 and the beta sheet midway along the interface between these elements.

The following figure supplement is available for figure 6:

**Figure supplement 1**. Relationship between nucleotide state and strain in the polypeptide backbone for a diverse array of kinesin X-ray structures.

## Discussion

The current work demonstrates that, while existing models of kinesin's ATP state on microtubules appear to be quite accurate (*Parke et al., 2010*; *Sindelar and Downing, 2010*; *Chang et al., 2013*; *Gigant et al., 2013*), the motor's nucleotide-free conformation exhibits a closed conformation of the switch loops that was not previously anticipated. This feature has lead to the identification of a novel allosteric pathway mediated by the 'linchpin' residue N255. The mechanism we propose can be summarized as follows (*Figure 8*). Upon encountering the microtubule, alpha tubulin engages N255 in order to stabilize the closed conformation of the switch loops. Here, the primary consequence of switch loop closure is to revise the mechanical linkages between kinesin's upper subdomain and the other two subdomains. In the detached ADP state of the motor, the upper subdomain forms a semi-rigid junction with the N-terminal subdomain (*Figure 8B*, top panels), via interactions between the open switch II loop and the P-loop. However, upon microtubule attachment the upper domain trades this junction for a new one with the lower subdomain, mediated by the closed switch loops (*Figure 8B*, bottom panels). The functional consequences of this change in subdomain connectivity are twofold. First, disengagement of the N-terminal subdomain from the upper subdomain allows the nucleotide cleft to relax into its open state and release ADP. Second, fixing the upper and lower subdomains

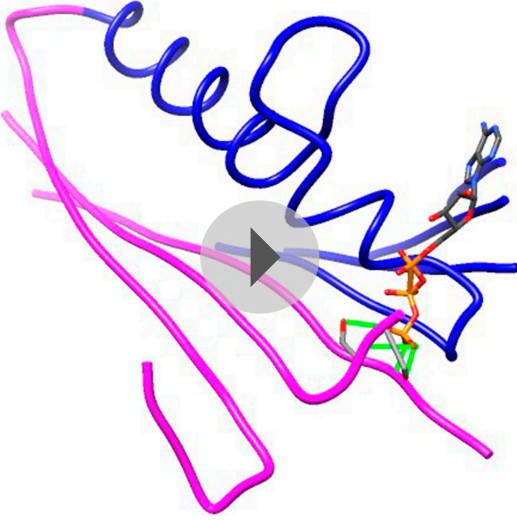

**Video 7**. Comparison of backbone distortions in the central beta sheet and α2 for our no-nucleotide and ATP-bound models of K349 kinesin, illustrated by a morphing animation. Compare with **Figure 6A,D**.

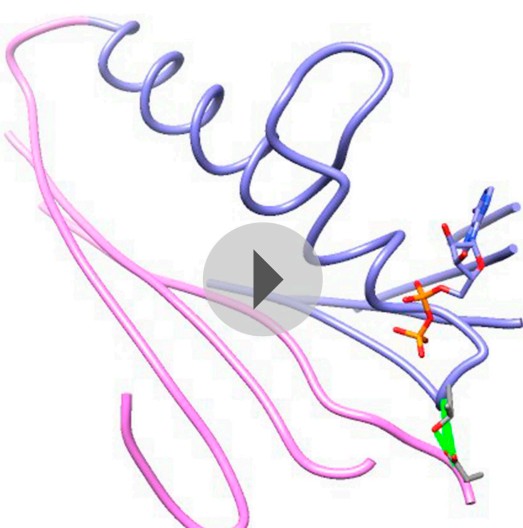

**Video 8**. Similar to **Video 7**, but substituting the X-ray structure of ADP-bound kinesin (PDB ID: 1MKJ) for the ATP-bound model. Compare with **Figure 6B,E**.

together introduces a well-defined allosteric pathway between the neck linker and the nucleotide cleft, so that closure of the nucleotide cleft by ATP becomes energetically coupled to docking of the neck linker.

An important distinction between the current scheme and earlier proposals for the kinesin ATPase cycle regards the structure of the nucleotide pocket. It is commonly proposed that ATP binding in kinesin would trigger a substantial rearrangement (or 'closure') of the universally conserved core segments of the switch I and II loops (DLAGSE, residues 231–236 and SSRSH, residues 201–205, respectively), accompanied by the formation of a catalytically activating salt bridge between E236 and R203 in switch I (**Sablin et al., 1996**; **Vale and Milligan, 2000**; **Kull and Endow, 2013**). The current results, however, suggest that these rearrangements are largely complete upon microtubule attachment even prior to ATP binding (**Figure 4**) and serve as an allosteric trigger for ADP release by detaching the switch II loop from the P-loop (**Figure 5**). Despite these rearrangements by switch I and switch II, the resulting no-nucleotide conformation of kinesin fails to attain its catalytically competent conformation, owing to relaxation of the nucleotide cleft into its 'open' state in which the P-loop withdraws from the switch loops. Only following ATP binding, when the nucleotide cleft re-closes, does kinesin attain its catalytically active configuration. These features of our structure models imply that the primary functional role of switch loop closure in kinesin is not to sense the gamma phosphate as was widely thought, but rather to dissociate ADP.

The behavior of the switch loops we infer for kinesin may be contrasted with that of the G proteins. In G proteins, binding of GTP typically transitions the switch loops from extended (and often partially disordered) conformations to ones that tightly coordinate the gamma phosphate, in order to modulate the binding affinity of various allosteric partner proteins (**Vetter and Wittinghofer, 2001**; **Wittinghofer and Vetter, 2011**). This behavior appears to be partially conserved in our microtubule-complexed models of kinesin, as reflected by a 'pincer-like' closure of loops L9 and L11 that contain switch I and switch II (respectively) and increased ordering of loop L9 that accompanies the ATP binding step (see **Figure 3C,F** and also **Figure 2—figure supplement 3A,B**). However, these changes in L9 and L11 are mainly limited to the non-conserved segments peripheral to the core switch loops and appear to have limited functional importance (see below), while the conserved switch regions remain locked in place via microtubule interactions of the N255 'linchpin' residue (**Figure 4**). Hence, in kinesin the switch loops appear to play a more passive role in gamma-phosphate sensing than in G proteins. Instead, our data indicate that the major allosteric transition triggered by ATP binding is closure of kinesin's nucleotide cleft, accompanied by large-scale distortion of the peptide backbone including the central beta sheet. This latter

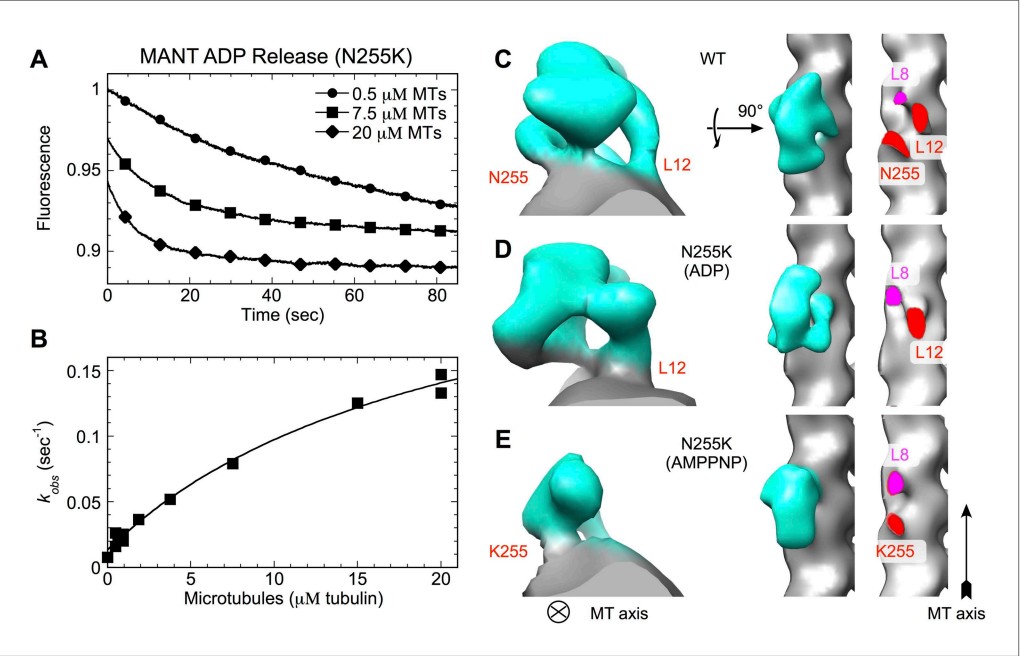

**Figure 7**. N255K uncoupling mutation compromises the microtubule interface, with accompanying orientational disorder of the motor domain. (**A**), (**B**) Microtubule-stimulated ADP release kinetics measurements for the N255K mutant of our K349 construct. Shown in (**A**) are individual time traces, while (**B**) plots the fitted rate constants as a function of microtubule concentration. The extrapolated maximum rate for microtubule-stimulated ADP release was $k_{max} = 0.26 \pm 0.04$ s$^{-1}$, with an apparent microtubule dissociation constant of $K_{0.5} = 21.3 \pm 6.3$ µM. (**C**–**E**) Cryo-EM reconstructions of microtubules decorated with wild-type K349 (**Sindelar and Downing, 2010**) and the N255K mutant. For the wild-type structure in (**C**), the ADP state was selected for viewing purposes but all nucleotide states have similar features at this resolution. Maps from reconstructions of 14-protofilament microtubules are shown; features in the corresponding 13-protofilament reconstructions were very similar (results not shown). All maps are low-pass filtered to 18 Å resolution, corresponding the estimated resolution of the N255K reconstructions. Supporting information describing cryo-EM analysis and 3D reconstruction of the N255K samples are shown in *Figure 7—figure supplement 1*.
The following figure supplement is available for figure 7:

**Figure supplement 1**. Analysis of cryo-EM images of N255K kinesin complexed with microtubules (see *Figure 7*).

mode of gamma-phosphate sensing, while not evident in the mechanism of G proteins, is characteristic of numerous other families of ATPase motors that utilize Walker motifs; these families include the ATP synthases (**Menz et al., 2001**), RNA and DNA helicases (**Yang, 2010**), as well as myosin (see below). The emergent theme of active site cleft closure in these diverse motor protein families may reflect the need to couple the energy of nucleotide binding and/or hydrolysis to large-scale domain motions necessary for mechanical energy transduction.

The structure models presented here contrast with a newly published cryo-EM study of the kinesin-microtubule complex by **Atherton et al. (2014)**. The structure models presented in that study are superficially similar to the ones identified in the current work, in that a 'closed' conformation of the switch loops was also identified in both no-nucleotide and ATP analog-bound states. However, the no-nucleotide conformation proposed by Atherton et al. seems to exhibit neither an open nucleotide cleft (as defined in the current work) nor twisting between the N-terminal and upper subdomains, both of which are signature traits of our no-nucleotide model. Consequently, the mechanistic scheme proposed by Atherton et al. is starkly different from the one summarized in *Figure 8B*. For example, in contrast to the ADP release mechanism proposed here, Atherton et al. suggest that closure of the switch I loop triggers ADP release by loosening interactions between this loop and a 'cap' of structured water molecules that coordinate the active-site magnesium cation (**Nitta et al., 2008**; **Atherton et al., 2014**). Notably, however, interactions between the switch I loop and the water

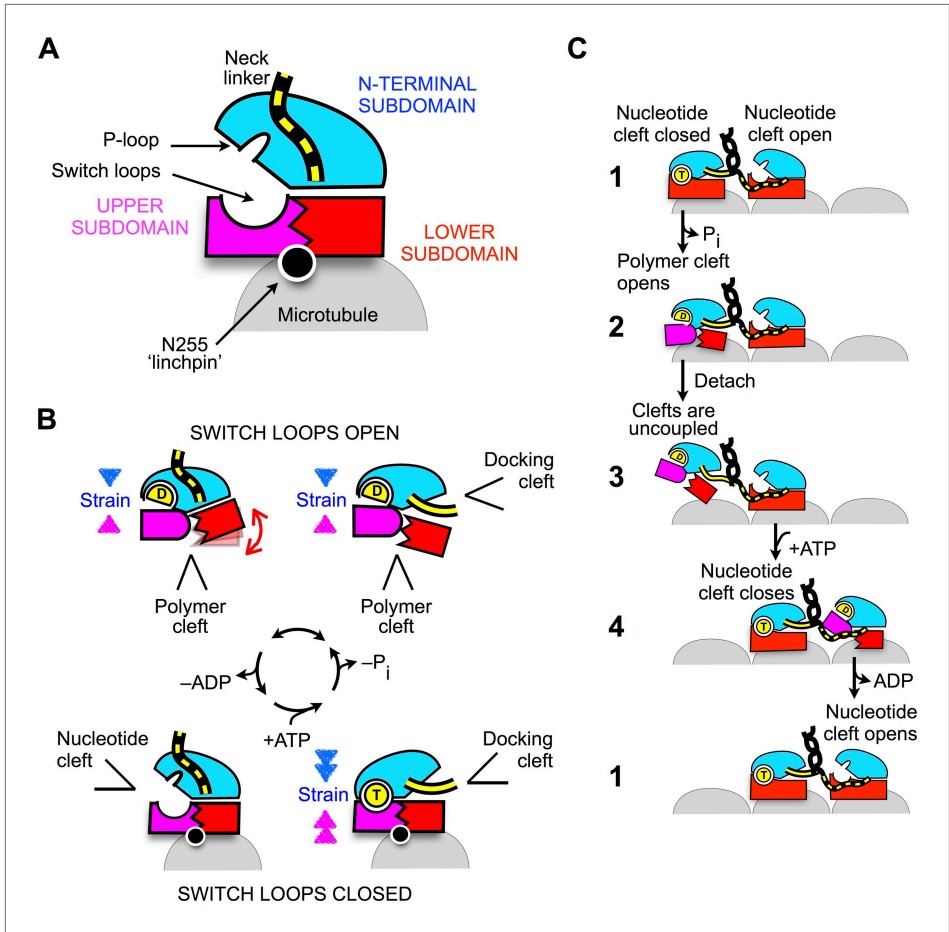

**Figure 8**. Schematic of proposed mechanisms of monomeric and dimeric conventional kinesin. (**A**) Abstracted cartoon that depicts kinesin's three principal subdomains and other key structural elements, including the 'linchpin' residue N255 (circle) and the neck linker (disordered state depicted as dashed yellow lines). (**B**) Schematic reaction depicting four principal structural intermediates in the ATPase cycle of monomeric kinesin. States of the nucleotide, polymer, and docking clefts are indicated. (**C**) Alternating cleft model for processive stepping by dimeric kinesin.

cap are quite variable in X-ray structures of ADP-bound kinesin and are usually indirect, mediated by at least one other water molecule. It is therefore not obvious that these kinds of interactions would be sufficiently specific to account for the substantial loss of ADP affinity that occurs with microtubule attachment (*Hackney, 1988*).

The transition that accompanies ATP binding in the Atherton et al. models is similarly distinct from the one described here. Atherton et al. propose that ATP binding actuates seesaw-like tilting of kinesin's central beta sheet via a pincer-like movement of the switch I loop and switch II loops toward each other. In our models, however, the switch loops mainly serve to anchor the lower half of the nucleotide cleft in place against the microtubule surface (via the closed hydrogen bond network, anchored by the N255 linchpin residue), while seesaw-like motion by the upper half of the nucleotide cleft (composed of the N-terminal subdomain) translates the P-loop down towards the switch loops. In our models, therefore, pincer-like motion of L9 and L11 can be likened to the action of a baseball glove: these loops come towards each other in order to 'catch' the gamma phosphate as the P-loop delivers ATP into them.

Several factors may underlie the disparities between the structures presented by Atherton et al. and ours, which appear to be mainly limited to the no-nucleotide state of kinesin. The improved resolution of the cryo-EM maps presented here (5–6 Å vs 6–7 Å in the Atherton et al. study) allows us to resolve the distinctive hooked shape of the P-loop in the no-nucleotide map, precisely defining its position with respect to the switch II loop and thus contributing to our identification of an open

conformation of the nucleotide cleft (compare *Figure 3* and *Figure 3—figure supplement 1* of the current work with *Figure 2* in Atherton et al.). Moreover, the simulation method used here to develop all-atom models places more emphasis on a chemically realistic simulation potential, by incorporating explicitly represented solvent molecules and keeping the magnitude of the perturbing cryo-EM potential as small as possible (See 'Materials and methods'). In contrast, Atherton et al. subdivided their starting models into smaller fragments, fit these as rigid bodies into their cryo-EM maps, and subsequently performed simulated annealing with an implicit solvent model. It therefore seems likely that lower resolution, different refinement methods, and also the use of additional a priori assumptions (i.e., fragment selection for the rigid-body fitting) in the methods of the Atherton et al. study may have impeded their identification of an open arrangement of the nucleotide cleft, in their no-nucleotide model.

## The connection between internal strain and nucleotide binding

It has increasingly been proposed that kinesin may harness the energy of twisting in its central beta sheet in some fashion during the ATPase cycle (*Hirose et al., 2006*; *Kull and Endow, 2013*; *Arora et al., 2014*). Speculations in this regard have mainly focused on the possibility, by analogy to myosin, that release of nucleotide initiates a much larger end-to-end twist in the sheet than is observed in comparisons of nucleotide-bound X-ray structures of the motor (*Gigant et al., 2013*; *Kull and Endow, 2013*; *Arora et al., 2014*). However, it has remained unclear how such twisting would relate to the complex interplay of affinities between kinesin and microtubules, ADP and ATP. The structure models derived in the current study indicate that in kinesin, the energy of nucleotide binding is captured by a related but distinct mode of distortion in the peptide backbone, involving coordinated rearrangements of the beta sheet and alpha 2 (*Figure 7*). As we have shown, this distortion is coupled with closure of the nucleotide cleft and can evidently be stabilized either by interactions between the P-loop and the open switch II loop (ADP state; *Figure 4F* and *Video 8*), or between the gamma phosphate and the closed switch loops (ATP state; *Figure 4E* and *Video 7*). This mechanism would thus allow microtubules to regulate kinesin's affinities for ADP and ATP by modulating the conformation of the switch loops, yielding an elegant structural basis for kinesin's nucleotide-binding behavior on and off microtubules.

## Structural basis for uncoupling of microtubule-activated functions by point mutations

A number of sites in the kinesin fold have been identified that uncouple the motor's nucleotide affinity from its microtubule affinity, but the structural basis of the uncoupling has remained unclear. The nucleotide-exchange mechanism proposed here predicts that the uncoupling mutations in the kinesin's switch regions would have differential effects on the kinetics of ADP release, depending on the nature of the affected site. For example, due to its apparent role in stabilizing the nucleotide cleft in the detached motor (see *Figure 4F*), perturbation of the E236 site would be expected to produce a comparatively large effect in kinesin's detached state, by destabilizing closure of the nucleotide cleft and increasing the ADP off rate. In contrast, mutation at the N255 site is not expected to produce a large effect in the detached motor but is expected to drastically attenuate microtubule-stimulated ADP release by compromising the 'linchpin' function of this residue. Both of these predictions are supported by kinetics measurements reported here and elsewhere, which reveal that the N255K mutation reduces the rate of microtubule-stimulated ADP release by up to 1000-fold, to near basal levels (*Auerbach and Johnson, 2005*) (*Figure 7A,B*), while the E236A mutation increases the ADP off-rate of detached kinesin by 100-fold (*Rice et al., 1999*) but does not significantly alter the rate of microtubule-stimulated ADP release. X-ray studies of the corresponding point mutants in KAR3 kinesin, co-complexed with ADP and magnesium, also support our interpretation. The structure of the KAR3 E631A mutant (equivalent to E236A) confirms that the mutated side chain can no longer interact strongly with the P-loop (*Yun et al., 2001*), supplying an explanation for the increased ADP off rate in the detached state of this mutant. In contrast, the structure of KAR3 N650K (equivalent to N255K) was found to be essentially identical to that of wild-type ADP complex (*Yun et al., 2001*), consistent with our proposal that the dominant functional effect originates in the microtubule complex rather than the detached state, for this latter mutant.

## 'Alternating cleft' model for processive movement by dimeric kinesins

The scheme in *Figure 8* also provides a framework for understanding how gating can be achieved between the motor domains of dimeric kinesin (*Figure 8C*). When both heads are attached to the

microtubule in their tight-binding states, the neck linker of the trailing head is constrained in an arrangement that favors docking, thus tending to keep the docking cleft open (*Figure 8C*, state 1). In contrast, the neck linker in the leading head is kept out of its docking cleft, thus favoring closure of the cleft. Thus, in the trailing head, tight coupling between the docking cleft and the nucleotide cleft (generated by the closed conformation of the switch loops) keeps the nucleotide cleft closed. Meanwhile, the same coupling pathway keeps the nucleotide cleft open in the leading head—weakening the nucleotide affinity and thus disfavoring the binding and/or hydrolysis of ATP.

Following hydrolysis and phosphate release in the trailing head (*Figure 8C*, state 2), however, strain generated at the active site in this head will no longer be supported by the nucleotide. At this point in the cycle, it appears that the neck linker would serve an important role as an allosteric modulator. If the neck linker were to be undocked following phosphate release, the trailing head would be free to revert to its previous conformation in the ATPase cycle (*Figure 8B*, lower left) and release ADP. However, if a forward step has occurred as depicted in states 1–2, the neck linker in the trailing head will continue to be held in a docked conformation and the only recourse to relieve strain at the active site is to open the switch loops, hence opening the polymer cleft. The resulting ADP state of the trailing head thus favors tight nucleotide binding and weak microtubule affinity-promoting detachment from (and discouraging subsequent reattachment to) the trailing microtubule site (*Figure 8C*, state 3). Importantly, binding of ATP in the leading head at this point in the cycle (*Figure 8C*, step 4) promotes a forward step by the trailing head through docking of the neck linker but does not prevent a backward step because neck linker docking is only weakly supported in the motor's ATP state (*Rice et al., 2003*). Thus, the properties just described for the ADP-bound trailing head are likely critical in preventing back-stepping, which is a key determinant of efficient and rapid motility.

Other structural schemes have been proposed in order to account for differences in nucleotide-binding and other properties between the leading and trailing heads of kinesin (*Hirokawa et al., 2009*; *Sindelar, 2011*). In such schemes the conformation of the neck linker was mainly thought to affect the nucleotide affinity by triggering rearrangement of the switch loops. However, our structural data do not support this type of mechanism, instead indicating that the switch loops are held in a closed conformation by the microtubule in both leading and trailing heads providing the basis for the gating scheme in *Figure 8C*.

## Insights on the relationship between the mechanisms of kinesin and myosin

The mechanism inferred here for kinesin reveals new parallels with the myosin molecular motor, which appears to be ancestrally related to kinesin (*Kull et al., 1998*) and shares very similar nucleotide-sensing 'switch' motifs (*Kull et al., 1996*; *Sablin et al., 1996*). While the structural basis of myosin's filament-attached power stroke is not well understood, it has been proposed that the corresponding lever-arm movement is coupled to progressive relaxation of the central beta sheet (corresponding to increasing degrees of twist) that would accompany catalysis and/or product release (*Coureux et al., 2004*). This proposal mirrors the sequence of events we infer for kinesin, where the magnitude of the peptide backbone distortion appears to be greatest in the ATP state, less for the ADP states, and least of all for the no-nucleotide conformation (*Figure 6C* and *Figure 6—figure supplement 1C-F*). A recent modeling study has also suggested that interactions with actin could lock myosin's switch I and switch II loops stably together while the P-loop moves with respect to them during the filament-attached power stroke (*Preller and Holmes, 2013*); this behavior would be directly analogous to the behavior of microtubule-attached kinesin in the currently proposed scheme.

An important functional difference between myosin and kinesin, however, is that these motors attach and detach from their partner filaments at different points in the ATPase cycle. In this light, it is worth noting that the 'linchpin' N255 residue in kinesin is conserved in myosin, but interacts natively with a conserved tyrosine found within the lower 50kD domain of myosin itself, rather than interacting with the filament as occurs with kinesin. Consequently, in myosin this asparagine holds the switch II loop near its 'closed' conformation even in the absence of actin, which could help explain why myosin, unlike kinesin, shows strong catalytic activity even in the absence of its partner filament (*Geeves and Holmes, 2005*). We anticipate that future studies of both kinesin and myosin will shed more light on the relationship between these intriguing molecular machines.

## Note added in proof

Concurrently with the current work, an X-ray crystal structure of no-nucleotide kinesin in complex with a non-polymerized tubulin dimer has been published (*Cao et al., 2014*). The structure of no-nucleotide kinesin reported in the X-ray study is in excellent agreement with model presented here (backbone RMSD for all visible residues is 1.28Å). Accordingly, the detailed interactions of kinesin and tubulin seen in the X-ray structure appear to be fully consistent with the models and mechanism presented in the current work.

# Materials and methods

## Protein expression and purification

The wild-type, monomeric human K349 construct was bacterially expressed and purified as described (*Kull et al., 1996*), and 15% (wt/vol) sucrose was added before snap freezing in liquid nitrogen and storing at −80°C. A plasmid for the mutant N255K construct was generated from the wild-type plasmid using the QuikChange site-directed mutagenesis kit (Agilent Technologies; Santa Clara, CA). Thawed K349 (either wild-type or N255K) was exchanged into EM buffer (25 mM PIPES, 25 mM KCl, 1 mM EGTA, 1 mM DTT) using three rounds of dilution and concentration in a Microcon ultracentrifugal filter (EMD Millipore; Billerica, MA). Microtubule batches were grown from 250 µg of lyophilized bovine brain tubulin (Cytoskeleton; Denver, CO), resuspended in 25 µl EM buffer and clarified (Beckman TLA 120.2, 100K RPM, 4°C) prior to incubation at 37°C. Taxol (2 mM in DMSO) was added to equimolar levels with tubulin after 10 min of incubation. Following ~45 min of polymerization, microtubules were brought to room temperature and a ~twofold excess of K349 was added prior to pelleting through a glycerol cushion (50 µl of EM buffer + 60% glycerol wt/vol + 200 µM taxol) in order to remove unbound motor and unpolymerized tubulin (20 min, Beckman TLA 120.2, 50K RPM, 24°C). The motor–microtubule complex was resuspended in ~10 µl of EM buffer plus 200 µM taxol.

## Mant-ADP release measurements

Stopped-flow measurements were performed at 298 K using an SF-300X (Indiana University) stopped-flow apparatus (KinTek Corp.; Austin, TX) equipped with a Xenon arc lamp (Hamamatsu; Japan). Kinetics of the interaction of mant-ADP with kinesin was measured by equilibrating a kinesin·mant-ADP (1:1) complex followed by rapid mixing with a high concentration of MgATP or varying MT concentrations plus MgATP to chase the mant-ADP from the active site as described (*Sadhu and Taylor, 1992*). Mant fluorescence was monitored over time, $I_{ex,max}$ = 356 nm, $I_{em,max}$ = 448 nm (400 nm long-pass filter).

## Cryo-EM sample preparation

Samples were plunge-frozen in liquid ethane, using Quantifoil holey carbon grids with 1 µm hole diameter, 1.5 µm spacing (Quantifoil Micro Tools GmbH; Germany). In order to optimize motor decoration on the microtubules, glow discharge was not applied to the grids and samples were diluted ~5–10× into distilled water (0.35 µl sample plus 3 µl d(H$_2$O)) on a piece of Parafilm prior to grid application, in order to compensate for evaporation that occurs prior to contact with liquid ethane (*Sindelar and Downing, 2007*). For the ADP, AMPPNP, and ADP•Al•F$_x$ conditions, the grid droplet mixture was supplemented by 2 mM nucleotide (2 mM ATP + 2 mM AlCl$_3$ + 10 mM NaF for the latter condition). Following our previously published protocol (*Sindelar and Downing, 2007*), most of the initially applied buffer on the grids was wicked away by touching the grid edgewise with a piece of filter paper (Whatman). The grids were then blotted completely, and after a 0.5–1 s delay, immersed into liquid ethane using a home-built plunge-freezing apparatus. Consistent with our prior experience (*Sindelar and Downing, 2007*), failure to dilute the sample buffer ~5–10× prior to plunge freezing led to poor and/or inconsistent kinesin decoration of the microtubules.

## Data collection and image processing

Micrographs of the microtubules decorated by the wild-type K349 construct were collected using the SerialEM package to collect data semi-automatically on 300 kV FEG-equipped electron microscopes (FEI F30 for the no-nucleotide data set, FEI Titan for the ADP•Al•Fx data set), using K2 direct electron detecting cameras (Gatan; Pleasanton, CA) in video mode. For bare microtubules, and microtubules decorated by the N255K construct, micrographs were collected using a TF-20 FEG-equipped instrument and a Gatan US4000 CCD detector. In all cases, the defocus was systematically varied from approximately 1 to 2.5 µM through the course of the data collection. The total dose was ~15 electrons/Angstrom

squared, distributed over 15 or 16 video frames. The no-nucleotide data set was collected at ~13K magnification using the camera's super-resolution mode and subsequently binned 2×, yielding 4K by 4K image dimensions with an effective pixel size of 1.99 Å. The ADP•Al•Fx data set was collected at approximately the same magnification as the no-nucleotide data set but in regular counting mode, resulting in 4K by 4K images with an effective pixel size of 2.097 Å.

After performing drift analysis (*Li et al., 2013*), whole-image video frames were aligned and averaged (K2 data only) before manually selecting overlapping boxed segments corresponding to individual microtubules. Defocuses and astigmatism parameters were estimated using the CTFFIND3 program (*Mindell and Grigorieff, 2003*). Single-particle analysis and 3D reconstruction was then performed as described (*Sindelar and Downing, 2010*), with some modifications. Initial reference models were generated by applying a parametric model of the microtubule (*Chrétien and Wade, 1991*) to PDB models of the no-nucleotide kinesin-microtubule complex (*Sindelar and Downing, 2007*), converting the atomic coordinates to EM density using the SPIDER package (*Frank et al., 1996*), and applying a low-pass filter with a 20 Å cutoff frequency. Multi-reference analysis performed using a range of different microtubule assemblies (12–15 protofilaments) established that 14-protofilament microtubules outnumbered 13-protofilament by approximately 3:1, with only marginal populations of other symmetry forms. Based on this determination, the 13- and 14-protofilament sub-populations were separately selected for further analysis.

Initial estimates of the in-plane rotation of individual segments were made using the Radon transform, as described (*Li et al., 2002*). Automated scripts using the SPIDER package (*Frank et al., 1996*) were used for semi-exhaustive searching of XY shifts and Euler angles (out-of-plane tilt range: +/−15°). Reference alignment was used to determine the position of the microtubule seam on a per-microtubule basis as described (*Sindelar and Downing, 2007*), as was the filament polarity, and then every segment was subjected to local refinement using the established seam orientation and polarity. Evidently due to disorder in the N255K kinesin head orientations, the correlation scores used to identify the seam orientation were far noisier for this mutant. Following initial refinement of the Euler angles and shifts for each box, the estimated position of each 8 nm repeat of the microtubule was mapped back onto the micrograph and a new stack of boxes was extracted for a final round of SPIDER refinement. Thus, the final data set included exactly one box for each 8 nm repeat identified in the selected microtubules. Subsequent structure refinement and 3D reconstruction was performed using the FREALIGN package (*Grigorieff, 2007*) with specific modifications for helical image processing (*Alushin et al., 2010*).

For the FREALIGN refinements, four rounds of refinement and reconstruction were initially performed, using successively higher resolution cutoffs for the refinement target (20 Å, 15 Å, 12 Å, 10 Å). For each reconstruction, 13- or 14-fold pseudo-helical symmetry was applied in order to transform all imaged asymmetric subunits onto a single 'good' protofilament, as previously described (*Sindelar and Downing, 2007*). Helical parameters were derived from the corresponding, canonical microtubule form (*Chrétien and Wade, 1991*) but adapted for the measured axial repeat distance in our data sets. The reconstructed 'good' protofilament (along with the remainder of the reconstruction) was subsequently replicated and transformed 13 or 14 times using the same symmetry parameters in order to generate protofilament models for the entire microtubule. The transformed 'good' protofilaments were subsequently selected by complementary wedge-shaped masks and summed in order to obtain the final asymmetric microtubule model. To reduce the influence of solvent noise in the refinement, a tight mask around the protein density was generated from a thresholded low-resolution version of the current map, and then smoothed with a soft Gaussian filter (~10 Å half-width). This mask was then applied to the reconstructed volume prior to the next round of refinement. The same mask was employed for FSC calculations.

Once the FREALIGN steps were completed, the resulting 3D map was then filtered to 20 Å and fed back into the SPIDER/FREALIGN pipeline, in order to reduce errors related to seam identification, polarity, and local searches. This additional step was omitted from the refinement of the N255K mutants. Following the second cycle of SPIDER/FREALIGN analysis, FSC calculations indicated that the resolution of the wild-type reconstructions was approximately 6 Å (0.143 criterion; *Figure 1—figure supplement 1A,B*), and no signs of higher-resolution features were apparent. We then subdivided the aligned video frames corresponding to each micrograph into summed groups of three (5 frames each), and re-extracted a new FREALIGN image stack from these sub-frames, resulting in a threefold larger number of stacked images. We then performed four more rounds of FREALIGN analysis with the new stack, using Euler angles and shifts determined previously as starting parameters. The resulting

reconstructions resolved individual beta strands and alpha helical pitch in some portions of the map (*Figure 1—figure supplement 1F,G*), although as noted in the main text the kinesin density was more poorly resolved and there was evidence of anisotropic blurring in the tubulin density (results not shown). The final resolution, obtained through FSC comparison of reconstructed half-data set volumes after applying a soft mask, was ~5 Å (0.143 criterion) for both nucleotide states (*Figure 1—figure supplement 1A,B*). For the final no-nucleotide reconstruction, a total of 10,029 8 nm repeats were used, for a total of ~140000 asymmetric units (14 protofilaments). A total of 2394 8 nm repeats, for a total of ~33,600 asymmetric units, were used for the final ADP•Al•F$_x$ reconstruction (14 protofilaments).

## Molecular dynamics flexible fitting calculations

In order to derive an atomic model for the microtubule complex of ATP-bound kinesin, we subjected the coordinates of tubulin and kinesin from the 4HNA structure to a hybrid all-atom molecular dynamics method (MDFF) in which cryo-EM density contributes a force field term (steering potential) designed to guide the simulation toward the experimentally observed structure (*Trabuco et al., 2008*). Version 1.91 of NAMD was used for all molecular dynamics calculations (*Phillips et al., 2005*). We used an explicit solvation model (TIP3P), because conformational instabilities were observed in trial simulations that used a less expensive continuum solvent model. All simulations were run using periodic boundary conditions, but no attempt was made to model the longitudinal or lateral interfaces of tubulin. While this omission likely produced artifacts near the boundaries between the simulated tubulin heterodimer and neighboring tubulin subunits, these boundaries are remote from the motor–microtubule interface. Moreover, the EM steering potential used in these simulations conformation strongly clamps the conformation of tubulin. Thus, our simulation setup is expected to minimize or eliminate the propagation of any boundary artifacts toward kinesin and its interface with tubulin, which is the focus of the current study.

To complete the starting model, twenty water molecules were manually placed within the kinesin nucleotide pocket at sites corresponding to crystallographic waters identified in a high-resolution (1.7 Å) X-ray structure of KIF4 kinesin co-complexed with AMPPNP (*Chang et al., 2013*). Hydrogen atoms were added to protein atoms and crystallographic water molecules using the 'guesscoord' command from the NAMD package (*Phillips et al., 2005*). The system was then placed in a TIP3P water box of dimensions 125 × 125 × 95 Å, and sodium and chloride ions were randomly added to this water box to a concentration of 50 µM, at a ratio that neutralized the charge of the system. After energy minimization (600 steps), the system was subjected to a three-phase equilibration, in which protein atoms were initially subjected to positional restraints that were then progressively released.

The equilibration simulations were performed in the NPT ensemble, using a Nosé–Hoover Langevin piston with a target pressure of 1 atm, a decay period of 200 fs and a time constant of 100 fs. The temperature was maintained at 310 K using a Langevin temperature bath with a time constant of 5 ps$^{-1}$. In the first equilibration phase, which ran for 50 ps, all protein and ligand atoms were restrained by a 20 kcal/mol/Å harmonic potential; the second equilibration step restrained protein backbone atoms only and the nucleotide ribose rings (50 ps); and the final step removed all harmonic constraints and was run for 600 ps. For all simulations, the CHARMM27 force field was used, with long-distance electrostatic interactions computed by the particle-mesh Ewald method and a non-bonded cutoff distance of 10 Å. Force field terms for GDP and GTP were adapted from ADP and ATP, respectively.

Following equilibration, we performed MDFF simulations, but with substantial modifications in order to reduce the possibility of overfitting, which has been shown to be a significant concern with this method (*Bai et al., 2013*). Importantly, we limited the influence of the EM steering potential to include only protein backbone atoms and nucleotide ligands. Thus, side chain atoms in our MDFF simulations are subjected exclusively to physics-based force field terms, which is reasonable because side chains were not visualized in our maps. Related to this point, as noted in the main text, a low-pass filter was used to remove signal beyond 6 Å resolution in the cryo-EM map, in order to reduce the level of contaminating noise prior to generating the EM steering potential. Limiting the EM steering potential to backbone atoms tends to moderate the contribution of the EM term in the simulated energy function, thus compensating for the limited number of experimental constraints available from our cryo-EM maps.

## Convergence of the fitted structures within the MDFF trajectories

In conventional applications of MDFF, the EM steering potential is introduced as a step function after an initial equilibration period, and the relative strength of the potential (compared with the physics-based terms in the molecular dynamics force field) is described by an adjustable parameter, ξ,

whose magnitude must be empirically determined. If the value of ξ is too low, it will not force a conformational transition within an achievable amount of simulation time; on the other hand, if the value is too high the molecular conformation becomes distorted ('overfitting'). However, a quantitative method for choosing an appropriate value for the ξ parameter has not been presented. To address this problem, we developed a modified MDFF protocol in which the value of ξ follows a linear ramp function, starting at zero and slowly increasing over the course of 10 nanoseconds. We then monitored the system for convergence by calculating the root mean squared deviation between the coordinates of the fitted structure and the starting model (*Figure 2—figure supplement 1*).

The resulting simulation trajectory exhibits a rapid conformational transition that occurs within the first 2 nanoseconds of the simulation, corresponding to relatively low values of the GSCALE parameter (GSCALE <0.15, compared with the 'typical' recommended value of 0.3 [*Trabuco et al., 2008*]). The RMSD signal was dominated by a large conformational change in tubulin, increasing to over 2.5 Å by the 2 ns time point (*Figure 2—figure supplement 1B*). This initial transition is followed by a gradual increase in the reported backbone RMSD values in both kinesin and tubulin that continues to the end of the simulation (GSCALE = 1.0). Inspection of the simulation trajectory reveals that the initial transition mainly consists of a straightening of the tubulin subunits from the conformation seen in 4HNA in order to accommodate the microtubule lattice. This transition is essentially complete after 1 ns of simulation time (see *Video 1*). In contrast to tubulin, the conformation of kinesin is largely unaffected during the first 2 ns of the MDFF simulation, and the backbone RMSD values (kinesin only) remain at approximately 1.0 Å with respect to the starting conformation for this period.

Subsequent to the first 2 ns of the MDFF simulation, the protein backbone progressively becomes disrupted beyond a level that can be justified by observable features in our EM density maps. A prominent example is loop L9 from kinesin, which encloses the nucleotide pocket and includes the switch I motif. In the starting model, a network of hydrogen bonds supports the structure of this loop; however, by the end of the simulation many of these interactions are lost as the loop is increasingly 'squeezed' into the corresponding density feature (results not shown). We note that the path of the protein backbone is not directly resolved this region of our density map, which means that the corresponding EM potential tends to uniformly pull backbone atoms toward the center of the loop feature (where density values are highest), and thus away from their true path. Disruption of L9 (and other secondary structure elements) in the latter parts of the ATP-state MDFF simulation is therefore likely due to over-fitting as the EM steering term increasingly predominates over the other energy terms in the simulated potential. We also note that the conformation of kinesin's N-terminal segment (residues 1–8), termed the neck cover strand (*Khalil et al., 2008*), is unstable even in early time points of the MDFF simulation, possibly due to weak density for both this element as well as the neck linker in the cryo-EM map. Limited occupancy of the docked neck linker is in fact expected under the experimental conditions used here (*Rice et al., 2003*). In order to minimize the presence of over-fitting artifacts in our final structure model, we selected an early time point (t = 1.2 ns) that immediately follows the initial transition, in order to represent the ATP-bound kinesin-microtubule complex (*Figure 2—figure supplement 1B*).

The stability of the resulting structure model was investigated using a follow-up simulation in which the EM steering potential was turned off but the conformation of tubulin was subjected to harmonic positional restraints (20 kcal/mol/Å) in order to maintain the straight conformation of tubulin found in the microtubule lattice. Only residues 1–380 of alpha and beta tubulin were included in the restraint potential, thus excluding the kinesin-binding regions (helices H11-H12). As shown in *Figure 2—figure supplement 1D*, the conformation of kinesin was relatively stable over the course of the 2 ns time period investigated, maintaining a backbone RMSD of ~1 Å from the starting structure. Signature hydrogen-bond interactions of the closed switch loop network (Y138, R203, E236, E250, N255) were maintained throughout this simulation, with the notable exception of the conserved salt bridge between R203 and E236, conserved in all X-ray structures where the switch loops are closed, which dissociated partway through the simulation (results not shown). This effect was also seen in unconstrained simulations of the 4HNA crystallographic model (results not shown), indicating a possible deficiency in the CHARMM27 energy function and/or initial structure model used here.

## MDFF model for no-nucleotide kinesin

In order to obtain an atomic model for kinesin's no-nucleotide state on microtubules, we repeated the above simulation protocol but replaced the target function with one derived from the no-nucleotide cryo-EM map. After modifying the 4HNA coordinates as described in the main text

(deletion of neck-linker residues and ATP), we equilibrated this system to allow water into the newly vacated nucleotide pocket, and then introduced an cryo-EM steering potential (linear ramp over 10 ns, as was done for the ATP state model) in order to bias the coordinates toward the conformation observed in our no-nucleotide map.

Similar to the ATP state trajectory, we observed a rapid conformational transition in the early phase of the MDFF simulation that was essentially complete by 1 ns (GSCALE = 0.2) (see *Video 2*). The behavior of tubulin in this trajectory, as reflected by the backbone RMSD from the starting structure, was also very similar to that seen in the ATP trajectory. The backbone RMSD values for kinesin, however, were much greater in the no-nucleotide trajectory, reflecting a significant conformational rearrangement. As described in the main text (*Figure 3*), the primary structural changes in kinesin were twisting of the N-terminal, upper, and lower subdomains with respect to each other, followed by separation of the P-loop from the switch II loop and attendant reorientation of the K91 side chain. The subdomain twisting movement was completed somewhat before the P-loop fully separated from the switch II loop, resulting in elongation of the P-loop as well as distortion of the closed switch loop network prior to the separation. After these elements had separated, however, they both 'snapped' back to conformations that resembled those seen in the starting model (*Video 3*). Similar to the behavior of tubulin in our simulations of both nucleotide states, the kinesin transition was largely complete by 1 ns, as reflected by the backbone RMSD from the starting structure (*Figure 2—figure supplement 1A*) as well as the observed interactions in the atomic model itself (*Video 3*). As with the ATP-state model, we therefore sought to avoid overfitting artifacts by selecting a trajectory frame immediately following the rapid transition (t = 1.4 ns) to represent the conformation of no-nucleotide, microtubule-complexed kinesin found in our maps. Generally, the agreement between the resulting atomic model and the cryo-EM map is excellent. The only obvious discrepancy between the map and the model is found in the vicinity of residues ~195–200 at the tip of L9, which forms part of the switch I response element. In this region, a substantial reduction in the average density of the no-nucleotide cryo-EM map indicates the presence of disorder. Consistent with this interpretation, we observed a corresponding increase in conformational fluctuations at the tip of L9 in a follow-up molecular dynamics simulation where the cryo-EM potential was turned off (See below), while the remainder of the active site remained firmly locked in place (*Figure 2—figure supplement 3A*).

We used the selected model from our no-nucleotide MDFF simulation to perform a 2 ns follow-up simulation where the EM steering potential was eliminated and the tubulin coordinates harmonically constrained, mirroring the protocol used for the ATP-state (above). This trajectory maintained the entire network of closed switch loop hydrogen bonds (Y138, R203, E236, E250, N255) found in the starting structure, including the salt bridge between R203 and E236; the active-site salt bridge formed between K91 and D231 in the starting model were also maintained throughout the trajectory (results not shown). These observations indicate that the no-nucleotide structure identified by our flexible fitting experiments is at least transiently stable. However, in this follow-up simulation kinesin exhibited substantially more flexibility (~1.5 Å RMSD) compared with the corresponding ATP-state trajectory (compare *Figure 2—figure supplement 1C,D*). This effect could be a consequence of the open conformation of the nucleotide cleft, and loss of attendant interactions between the switch II and the P-loop, seen in the no-nucleotide state structure.

## Molecular Graphics

All figures and videos were rendered with UCSF Chimera (*Pettersen et al., 2004*).

## Accession codes

Cryo-EM maps and coordinates for no-nucleotide and ADP•Al•Fx states of the kinesin-microtubule have been deposited in the EMDB (accession codes 6187 and 6188, respectively) and PDB (ID codes 3J8X and 3J8Y, respectively).

## Acknowledgements

We gratefully acknowledge Daniel Iwamoto, Daifei Liu, and Tony Schramm for their contributions in generating and expressing the N255K construct; Angelo Morales for assistance with sample preparation; and Sarah Rice for the generous donation of purified wild-type K349 protein. We thank staff in the Yale CryoEM facility and High-Performance Computing facility for their maintenance of these facilities; Eva Nogales and Bob Glaeser for access to the Berkeley EM facility; and Niko Grigorieff

and Daniela Nicastro for access to the Brandeis EM facility. We thank Joe Howard, Henry Shuman, and David Schatz for critical reading and constructive comments on the manuscript.

## Additional information

### Funding

| Funder | Grant reference number | Author |
| --- | --- | --- |
| American Cancer Society | Institutional Research Grant for New Investigators, ACS-IRG 58-012-55 | Zhiguo Shang, Kaifeng Zhou, Charles V Sindelar |
| National Institutes of Health | R01 GM 110530-01 | Zhiguo Shang, Kaifeng Zhou, Charles V Sindelar |

The funders had no role in study design, data collection and interpretation, or the decision to submit the work for publication.

### Author contributions

ZS, Acquisition of data, Analysis and interpretation of data, Drafting or revising the article; KZ, Performed sample preparation, electron microscopy experiments and data processing; CX, RC, Assisted with electron microscopic data collection; JCC, CVS, Conception and design, Acquisition of data, Analysis and interpretation of data, Drafting or revising the article

## Additional files

### Major datasets

The following datasets were generated:

| Author(s) | Year | Dataset title | Dataset ID and/or URL | Database, license, and accessibility information |
| --- | --- | --- | --- | --- |
| Shang Z, Zhou K, Xu C, Csencsits R, Cochran JC, Sindelar CV | 2014 | High-resolution structure of no-nucleotide kinesin on microtubules | http://www.pdb.org/pdb/search/structidSearch.do?structureId=3J8X | Publicly available at RCSB Protein Data Bank. |
| Shang Z, Zhou K, Xu C, Csencsits R, Cochran JC, Sindelar CV | 2014 | High-resolution structure of ATP analog-bound kinesin on microtubules | http://www.pdb.org/pdb/search/structidSearch.do?structureId=3J8Y | Publicly available at RCSB Protein Data Bank. |
| Shang Z,Zhou K, Xu C, Csencsits R, Cochran JC, Sindelar CV | 2014 | Microtubule decorated with monomeric human kinesin (K349 construct) having an empty nucleotide pocket | http://www.ebi.ac.uk/pdbe/entry/EMD-6187 | Publicly available at Electron Microscopy Data Bank. |
| Shang Z, Zhou K, Xu C, Csencsits R, Cochran JC, Sindelar CV | 2014 | High-resolution structures of kinesin on microtubules provide a basis for nucleotide-gated force generation | http://www.ebi.ac.uk/pdbe/entry/EMD-6188 | Publicly available at Electron Microscopy Data Bank. |

The following previously published datasets were used:

| Author(s) | Year | Dataset title | Dataset ID and/or URL | Database, license, and accessibility information |
| --- | --- | --- | --- | --- |
| Sindelar CV, Budny MJ, Rice S, Naber N, Fletterick R, Cooke R | 2002 | Human Kinesin Motor Domain With Docked Neck Linker | http://www.pdb.org/pdb/explore/explore.do?structureId=1MKJ | Publicly available at RCSB Protein Data Bank. |
| Kull FJ, Sablin EP, Lau R, Fletterick RJ, Vale RD | 1996 | Human ubiquitous kinesin motor domain | http://www.pdb.org/pdb/explore/explore.do?structureId=1BG2 | Publicly available at RCSB Protein Data Bank. |
| Arora K, Talje L, Asenjo AB, Andersen P, Atchia K, Joshi M, Sosa H, Allingham JS, Kwok BH | 2014 | Crystal structure of the mouse KIF14 motor domain | http://www.pdb.org/pdb/explore/explore.do?structureId=4OZQ | Publicly available at RCSB Protein Data Bank. |

| | | | | |
|---|---|---|---|---|
| Kikkawa M, Sablin EP, Okada Y, Yajima H, Fletterick RJ, Hirokawa N | 2001 | Crystal structure of the KIF1A motor domain complexed with Mg-ADP | http://www.pdb.org/pdb/explore/explore.do?structureId=1I5S | Publicly available at RCSB Protein Data Bank. |
| Coureux P-D, Sweeney HL, Houdusse A | 2004 | Crystal structure of myosin V motor domain -nucleotide-free | http://www.pdb.org/pdb/explore/explore.do?structureId=1W8J | Publicly available at RCSB Protein Data Bank. |
| Coureux P-D, Sweeney HL, Houdusse A | 2004 | Crystal structure of myosin V motor with essential light chain + ADP-BeFx - near rigor | http://www.pdb.org/pdb/explore/explore.do?structureId=1W7J | Publicly available at RCSB Protein Data Bank. |
| Menz RI, Walker JE, Leslie AGW | 2001 | (ADP.ALF4)2(ADP.SO4) Bovine F1-ATPase (all three catalytic sites occupied) | http://www.pdb.org/pdb/explore/explore.do?structureId=1H8E | Publicly available at RCSB Protein Data Bank. |
| Turner J, Anderson R, Guo J, Beraud C, Fletterick R, Sakowicz R | 2001 | Crystal Structure of the Mitotic Kinesin Eg5 in Complex with Mg-ADP | http://www.pdb.org/pdb/explore/explore.do?structureId=1II6 | Publicly available at RCSB Protein Data Bank. |
| Yun M, Zhang X, Park CG, Park HW, Endow SA | 2001 | Crystal structures of mutants reveal a signalling pathway for activation of the kinesin motor ATPase | http://www.pdb.org/pdb/explore/explore.do?structureId=1F9V | Publicly available at RCSB Protein Data Bank. |
| Parke CL, Wojcik EJ, Kim S, Worthylake DK | 2010 | Human kinesin Eg5 motor domain in complex with AMPPNP and Mg2+ | http://www.pdb.org/pdb/explore/explore.do?structureId=3HQD | Publicly available at RCSB Protein Data Bank. |
| Chang Q, Nitta R, Inoue S, Hirokawa N | 2013 | Crystal Structure of the Kif4 Motor Domain Complexed With Mg-AMPPNP | http://www.pdb.org/pdb/explore/explore.do?structureId=3ZFC | Publicly available at RCSB Protein Data Bank. |
| Gigant B, Wang W, Dreier B, Jiang Q, Pecqueur L, Pluckthun A, Wang C, Knossow M | 2013 | Kinesin motor domain in the ADP-MG-ALFX state in complex with tubulin and a DARPIN | http://www.pdb.org/pdb/explore/explore.do?structureId=4HNA | Publicly available at RCSB Protein Data Bank. |
| Yun M, Zhang X, Park CG, Park HW, Endow SA | 2001 | Crystal structures of kinesin mutants reveal a signalling pathway for activation of the motor ATPase | http://www.pdb.org/pdb/explore/explore.do?structureId=1F9T | Publicly available at RCSB Protein Data Bank. |

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
