## [Decision Letter]

Thank you for sending your work entitled “High-resolution structures of the
kinesin-microtubule complex reveal the basis of nucleotide-gated force
generation” for consideration at *eLife*. Your article has been
favorably evaluated by John Kuriyan (Senior editor), a Reviewing editor, and 2
reviewers.

As you can see below, both reviewers are positive about your work, and I enclose their
reviews for your consideration. Before agreeing to final publication, we would like you
to address the point 2 of reviewer 1 by modifying the Discussion.

*Reviewer #1*:

This work confirms the results of the recent Moores' paper at somewhat higher
resolution and goes on to provide some interesting new insights into the
nucleotide-dependent interaction of kinesin-1 with MTs. The data are excellent but I
think the text can be improved; shorter would surely be better.

1) It is a significant achievement to have obtained 6A resolution in the kinesin
subunits, which are notoriously mobile relative to the MT. Thus the crystal structure of
ATP-kinesin complexed with a tubulin heterodimer has now been proved to adequately
represent kinesin bound to a MT. This suggests that the crystal structure of apo-kinesin
complexed with tubulin will be equally relevant and avoid the daunting task of pushing
the EM data to even higher resolution.

2) I am not convinced by the claim that kinesin has diverged from the basic G protein
mechanism. The loops around the empty active site may well be firmly closed in the
frozen state that is being observed but they are probably free to open and close due to
thermal fluctuations at working temperatures. Obviously, this freedom is lost when
tilting of the domain clamps the surfaces together and the site becomes truly closed.
Presumably the domain is free to tilt except when ATP is present, holding the surfaces
together or when the neck linker is wedged in position.

3) The clashing of N255 is interesting whatever one's thoughts about the
mechanism.

4) The discussion about how the dimer is coordinated is very convincing.

*Reviewer #*2:

This paper presents results of a cryo-electron microscopy study of kinesin-decorated
microtubules in two nucleotide states. It advances our understanding of the mechanics of
kinesin by providing improved data on the conformations in these states and insights on
the transitions between them. This paper follows the recent publication of a closely
related study by Atherton et al. in *eLife* a few weeks ago. These papers
used essentially the same techniques and reach nearly the same resolution, which sets
the limit to how much can be interpreted from the EM data. This is an incremental
improvement in resolution over previously published results, but in both cases new
features and mechanisms can be identified from the resultant density maps and
differences among them. The nominal resolution in Atherton et al is reported as 6-7
Å and in the current work 5-6 Å. Although both sets of density maps were
filtered to around 6 Å, and the maps are presented at different threshold levels,
it does appear that Shang et al do have sufficiently better resolution to justify their
confidence in interpreting some important loop conformations. The authors were also able
to do a more extensive fitting of atomic models based on their maps. The two papers do
come to some of the same conclusions, but differ in significant ways with regard to the
loops and the core beta sheet, and these differences have substantial impact of
mechanistic models. Shang et al suggests a more complicated domain structure than has
been seen before, with identification of a “polymer cleft” in the
microtubule interface region. The beta sheet appears to be strained rather than just
twisted, which may relate to the mystery of where the energy of ATP hydrolysis goes.
Perhaps most importantly, the new maps indicate that movement of the P-loop is more
significant than opening or closing of the switch-I/switch-II region for nucleotide
binding and release.

Altogether this is a substantive and interesting advance that appears to answer a number
of puzzles about kinesin's action. The conclusions are well supported by the
experimental results, which include mutation of the critical N255 residue as well as
application of molecular dynamics in building the atomic models. The instability of the
simulations over long times could be a concern, but does not bother me in the present
context. The correspondence with myosin activity in this paper extends ideas presented
earlier by Dr. Sindelar and provides further support for the functional models presented
here, although further work at higher resolution will be required to validate these
proposals. In terms of validation tests suggested recently by Henderson and others, I am
comfortable that this work meets standards of acceptance.

---

## [Author Response]

*I am not convinced by the claim that kinesin has diverged from the basic G
protein mechanism. The loops around the empty active site may well be firmly closed
in the frozen state that is being observed but they are probably free to open and
close due to thermal fluctuations at working temperatures. Obviously, this freedom is
lost when tilting of the domain clamps the surfaces together and the site becomes
truly closed. Presumably the domain is free to tilt except when ATP is present,
holding the surfaces together or when the neck linker is wedged in
position*.

We agree with Reviewer #1 that the switch loops will be subject to greater thermal
fluctuations in the no-nucleotide state, compared with the ATP state, despite our
labeling them as ‘closed’; this was particularly evident in our free
simulations of the no-nucleotide structure, which identified large oscillations in the
conformation of L9, as is described in Figure 2—figure supplement 3.

Moreover, the switch loops clearly move closer in to the nucleotide pocket upon ATP
binding, in a ‘pincer-like’ movement as was described here and in Atherton
et al. Thus, even in our data, kinesin’s switch loops retain a ‘G-protein
like’ character which could even be greater at room temperature, as compared to
the frozen specimens imaged here. We have removed the original discussion paragraph on
this topic and added two new paragraphs. We soften the claim of
‘divergence’ from G-proteins, and instead emphasize the new parallels
identified with other Walker-containing motors such as DNA/RNA helicases, F1-Atpase and
myosin where a cleft is apparently used to store energy in response to nucleotide
binding and/or hydrolysis.